# A giant virus forms a specialized subcellular environment within its amoeba host for efficient translation

Ruixuan Zhang [1,7], Lotte Mayer [2,7], Hiroyuki Hikida [1,6], Yuichi Shichino [3,4], Mari Mito [3], Anouk Willemsen [2], Shintaro Iwasaki [3,5] ✉ & Hiroyuki Ogata [1] ✉

Many eukaryotic viruses, including amoeba-infecting mimiviruses, have codon usage that deviates from their hosts. However, codon usage patterns that align with the cellular tRNA pool enable efficient translation. How these viruses cope with the mismatch between tRNA supply and demand is unclear. Here we show that Acanthamoeba polyphaga mimivirus (APMV) generates a subcellular area to translate viral mRNAs. tRNA sequencing showed that the tRNA pool was not substantially altered during the infection, even though the virus encodes tRNA genes. Using in situ labelling, we found that viral mRNAs and newly synthesized proteins were localized in the periphery of the viral factory, suggesting that APMV creates a discrete subcellular environment to facilitate translation. Frequently used codons in viral mRNAs had higher tRNA accessibility than the same type of codons in amoeba mRNAs. Our data show how local translation assists the virus in overcoming the mismatch between tRNA supply and demand.

Codon usage in genes is unique in each genome. Even for synonymous codons that encode the same amino acid, the codon usage frequency varies. Preferentially used codons are associated with high tRNA copy numbers in fast-growing microorganisms such as *Escherichia coli*[1]. Recent studies highlighted the importance of synonymous codon usage in mRNA stability[2,3]. Therefore, the concordance of tRNA supply and demand is crucial for ensuring efficient gene expression.

Giant viruses are characterized by huge genomes, a large number of genes for reprogramming host cell metabolism and large virions[4–7]. Infection by giant viruses induces a rapid shift in the host cell transcriptome, characterized by large occupancy of viral mRNAs[8–10]. However, the codon usage of some giant viruses is poorly adapted to the tRNA pool of the host[11,12]. For example, APMV—the first virus isolate of family

*Mimiviridae*—has an AT-rich genome (G+C content of 28%), whereas the genome of its host *Acanthamoeba castellanii* is GC-rich (G+C content 58%)[13,14]. Consequently, the APMV codon usage varies extensively from the codon usage of *A. castellanii*[11,13]. This raises the question of whether translation of APMV mRNAs is negatively impacted by the codon usage or whether APMV alleviates the unfavourable translation condition during infection.

Several observations and hypotheses have been made related to the codon usage discrepancy. Genomic analysis found that members of the *Mimiviridae* encode more translation-related genes, including translation factors, tRNAs and aminoacyl-tRNA synthetases[4–7], than other clades of giant viruses. A transcriptomic study reported that the late-expressed genes of APMV exhibit a slight adaptation to the host

[1]Bioinformatics Center, Institute for Chemical Research, Kyoto University, Uji, Japan. [2]Centre for Microbiology and Environmental Systems Science, Division of Microbial Ecology, University of Vienna, Vienna, Austria. [3]RNA Systems Biochemistry Laboratory, Pioneering Research Institute, RIKEN, Wako, Japan. [4]Department of RNA Biochemistry, Institute of Medicine, University of Tsukuba, Tsukuba, Japan. [5]Department of Computational Biology and Medical Sciences, Graduate School of Frontier Sciences, The University of Tokyo, Kashiwa, Japan. [6]Present address: Research Center for Biosafety, Laboratory Animal and Pathogen Bank, National Institute of Infectious Diseases, Japan Institute for Health Security, Shinjuku, Japan. [7]These authors contributed equally: Ruixuan Zhang, Lotte Mayer. ✉e-mail: shintaro.iwasaki@riken.jp; ogata@kuicr.kyoto-u.ac.jp

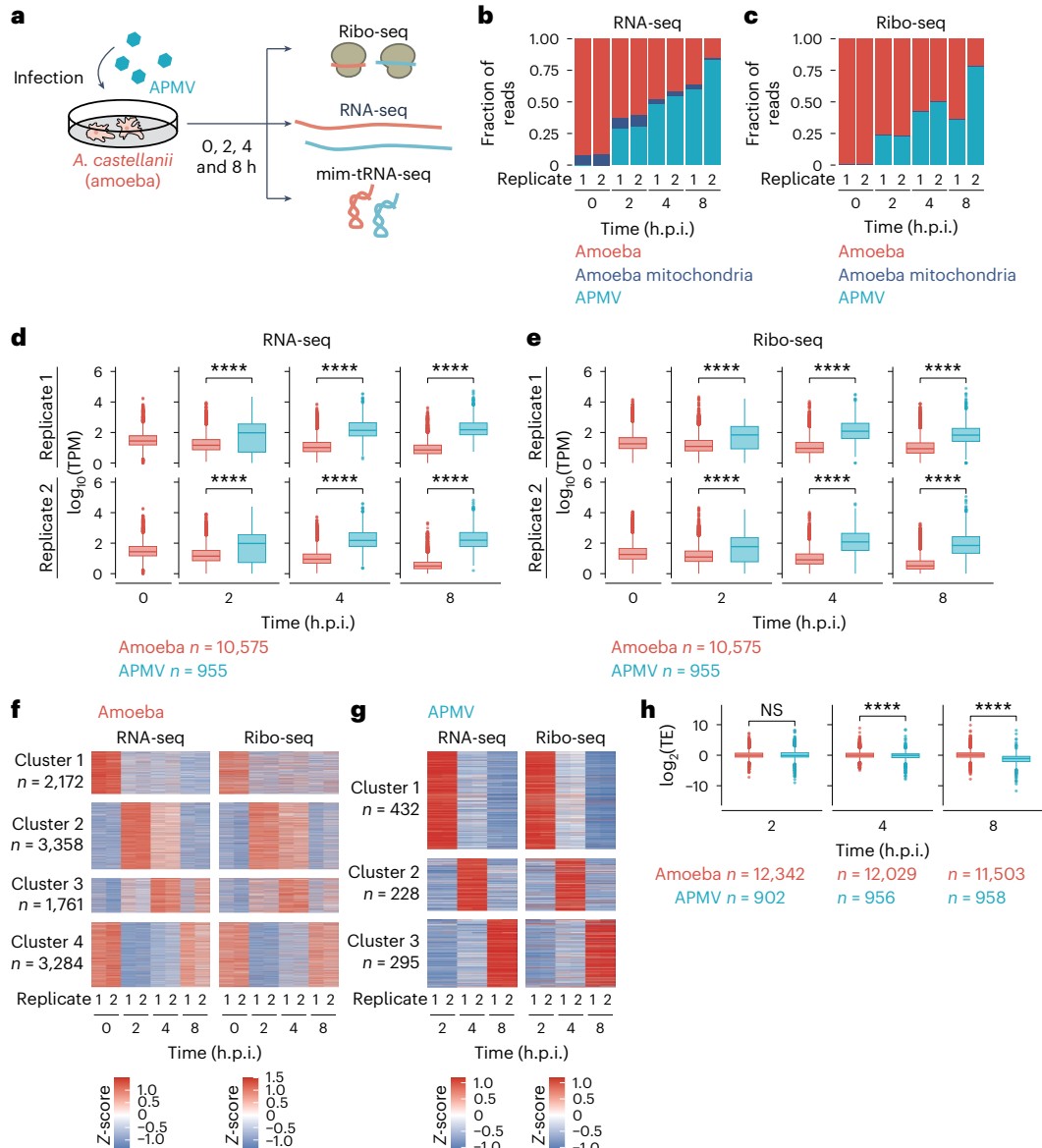

**Fig. 1 | Global impacts of APMV infection on RNA abundance and translation.**
**a**, Schematic diagram of the experimental design. Cells were infected with APMV for 0 (mock infection), 2, 4 or 8 h and used for RNA-seq, Ribo-seq and mim-tRNA-seq. **b**,**c**, Fraction of reads that aligned to indicated genomes by RNA-seq (**b**) and Ribo-seq (**c**) analyses at different infection stages. **d**,**e**, Box plots of transcripts per kilobase million (TPM) for amoeba and APMV mRNAs in the RNA-seq (**d**) and Ribo-seq (**e**) data. **f**,**g**, Heat maps of the Z-score normalized TPM in the RNA-seq and Ribo-seq data for amoeba (**f**) and APMV (**g**) mRNAs at different infection stages. Colour scales indicate the Z-scores. **h**, Box plots of translation efficiency (TE) of amoeba and APMV mRNAs at different infection stages. In all box plots, the median (centre line), upper/lower quartiles (box limits), 1.5× interquartile range (whiskers) and outliers (points) are shown. The P values were calculated using one-sided Wilcoxon rank sum test. NS, not significant; *P < 0.05; **P < 0.01; ***P < 0.001; ****P < 0.0001.

tRNA pool[11]. These findings suggest that translation-related fitness has shaped viral genes and genome evolution. However, the actual translation dynamics in infected cells have never been explored so far. Moreover, APMV forms an organelle-like structure known as a viral factory upon infection for viral DNA replication and transcription[15,16]. Ribosomes and some translation factors have been observed to localize around the viral factory[17,18]. How this spatial organization within infected cells relates to the observed patterns of genome evolution is still unknown.

To characterize the translation landscape of mimiviruses, we monitored changes in the transcriptome, tRNAs and translatome during viral infection by RNA sequencing (RNA-seq), optimized tRNA-seq[19] and ribosome profiling (Ribo-seq)[20,21]. Ribo-seq captures ribosome-protected fragments of mRNA, so-called 'ribosome footprints', generated by

RNase treatment, enabling high-resolution measurements of translation dynamics. Our results indicate that the overall composition of the global tRNA pool was not significantly altered during APMV infection, with limited contribution of tRNAs encoded in the viral genome. At the early infection stage, the translation efficiency of the viral mRNAs was comparable to that of the host mRNAs despite using rare codons. Even at the later infection stages, the rare codons did not cause the ribosome to pause on viral mRNAs; rather, the accessibility to tRNAs for codons ending with A or U was different between viral and host mRNAs. The microscopic analysis using fluorescence in situ hybridization (FISH) and fluorescent non-canonical amino acid tagging (FUNCAT)[21] showed that the spatial organization within infected cells indicated co-localization of viral mRNAs, host ribosomal RNAs and newly synthesized proteins around the viral factory. Our results

suggest that APMV created a subcellular heterogeneity for efficient translation of the viral genes.

## Results

### Imbalance of codon usage between APMV and the amoeba host

We characterized the dissimilarity in codon usage between APMV and the host *A. castellanii* (hereafter referred to as amoeba). Consistent with previous genome analyses[13,14], the host and viral genes had distinct G+C content (Extended Data Fig. 1a) and codon usage (Extended Data Fig. 1b). APMV mRNAs predominantly use AU-rich codons, such as AAA, AAU and AUU, whereas AU-rich codons were much less common in amoeba mRNAs. The tRNA gene copy number of amoeba was positively correlated with the amoeba codon usage (Extended Data Fig. 1c), whereas it showed no such correlation with the APMV codon usage (Extended Data Fig. 1d).

### Impact of APMV infection on RNA abundance and translation

To investigate the dynamics of transcription and translation of viral and host genes, we performed RNA-seq and Ribo-seq in APMV-infected amoeba cells at 0, 2, 4 and 8 h post-infection (h.p.i.) (Fig. 1a and Supplementary Table 1). We avoided pre-treatment with cycloheximide in the media, as it may induce bias through aberrant ribosome pausing in a context-dependent manner[22–25]. Instead, the post-lysis approach (that is, cycloheximide in cell lysis buffer), which is a current standard[26–28], was used. Consistent with an earlier report[29], the footprints showed two length peaks at 21–22 and 29–30 nucleotides (nt) (Extended Data Fig. 2a,b). The footprint counts along genes showed a clear 3 nt periodicity for both long (29–30 nt) (Extended Data Fig. 2c–f) and short (21–22 nt) footprints (Extended Data Fig. 2g–j). Discrete Fourier transform analysis further ensured the periodicity (Extended Data Fig. 3a). Our data had high reproducibility for read mapping for both the host and viral genes (Extended Data Fig. 3b–e). These features indicate the robust detection of footprints generated by ribosomes translating mRNAs for both host and virus.

The RNA-seq reads from viral mRNAs increased incrementally along with the viral infection time and were dominant at the late time points (up to approximately 62% and 85% for each replicate at 8 h.p.i.) (Fig. 1b). Footprints from viral mRNAs also increased and occupied a large proportion of the library at 8 h.p.i. (approximately 40% and 81% for each replicate) (Fig. 1c). We also observed a general increase of short footprints during virus infection (Extended Data Fig. 3f,g; see Methods for definition).

The average expression levels of viral mRNAs were higher than those of amoeba mRNAs at all time points (Fig. 1d). The expression levels of individual viral mRNAs increased gradually, whereas those of host mRNAs decreased throughout infection (Fig. 1d). Translation activity, evaluated by Ribo-seq, followed the same trends as that obtained by RNA-seq (Fig. 1e).

The host genes were classified into four clusters based on their mRNA levels from RNA-seq (Fig. 1f). To characterize the functions of genes in each cluster, we selected host genes that had high variance in their expression and performed a functional enrichment analysis using the Kyoto Encyclopedia of Genes and Genomes (KEGG) (Extended Data Fig. 3h and Supplementary Table 2). Genes in cluster 1, which showed an expression shutdown by 2 h.p.i., were enriched in energy metabolism, amino acid metabolism and motor proteins. Ribosomal genes had various expression patterns and were enriched in clusters 1, 3 and 4. tRNA and protein processing functions were enriched in clusters 2 and 3. Viral mRNAs were classified into three expression clusters (Fig. 1g), which largely correspond to the early, intermediate and late genes reported previously[8]. The translation levels of host and viral mRNAs in individual clusters showed dynamics consistent with the transcription patterns, suggesting that the abundance of mRNAs is a major determinant of the level of translation (Fig. 1f, g).

To assess the density of ribosomes on mRNAs, we calculated the translation efficiency as the ratio of the amounts of footprints to the amounts of mRNAs. At 0 and 2 h.p.i., the median translation efficiency values were 0.991 and 0.992 for host mRNAs, respectively, and 0.935 (2 h.p.i.) for viral mRNAs, indicating no significant difference between the translation efficiency of host and viral mRNAs (Fig. 1h). However, viral translation efficiency dropped and was significantly lower than that of host translation efficiency at later time points; at 4 and 8 h.p.i., the median translation efficiency values were 0.989 and 1.010 for host mRNAs, and 0.956 and 0.468 for viral mRNAs (Fig. 1h). This result suggests that host mRNAs were twice as likely to bind to ribosomes as viral mRNAs at 8 h.p.i., indicating the reduced accessibility to ribosomes for viral mRNAs at later time points.

### Translation of viral mRNAs is not associated with ribosome pausing

Given the deviated codon usage of APMV mRNAs, we investigated whether the rare codons on viral mRNAs hampered virus mRNA translation. As APMV genes are biased to codons ending with A or U (hereafter AU3 codons) and host genes are biased to codons ending with G or C (hereafter GC3 codons) (Extended Data Fig. 1a), we investigated the translation elongation speed on codons from viral and host mRNAs by calculating the A-site ribosome occupancy. No significant difference was found in ribosome occupancy at the A-site between AU3 and GC3 codons for amoeba (Fig. 2a), and this balanced elongation was maintained in viral mRNAs (Fig. 2b).

Next, we focused on individual codon positions to determine whether ribosomes tended to pause on either viral or host mRNAs. Codon positions with relatively high ribosome occupancy were defined as pause sites (Fig. 2c and Methods). We found that viral mRNAs had a lower tendency of ribosome pausing than host mRNAs did (Fig. 2d), indicating that ribosome traversal on viral mRNAs was smooth, even with the obvious codon usage conflict (Extended Data Fig. 1). This result is robust under different cut-offs for putative pausing site prediction (Extended Data Fig. 4a).

### The tRNA pool was stable during viral infection

Because APMV encodes tRNA genes (three Leu (two TTAs, one TTG), Trp (TGG), Cys (TGC) and His (CAC))[13], we hypothesized that alteration of the cellular tRNA pool facilitated ribosome traversal along viral mRNAs. To assess the abundance of tRNA species, we applied modification-induced misincorporation tRNA sequencing (mim-tRNA-seq)[19]. As tested in an earlier study[19], we also evaluated the salt concentration, pH, temperature and reverse transcription enzymes for optimal cDNA synthesis (Extended Data Fig. 5a,b). Ultimately, we selected the most efficient condition (with the thermostable group II intron reverse transcriptase, TGIRT-III, at 49 °C for 16 h) for downstream tRNA-seq. The tRNA supply ($W$-score; see Methods) showed a significant negative correlation with ribosome occupancy on codons in growing amoeba (Extended Data Fig. 5c), further ensuring that our Ribo-seq reflects ribosome traversal speed in cells.

The mim-tRNA-seq showed that the expression of tRNAs encoded in the amoeba genome remained at a high level (>87% in every library), whereas the expression of the six tRNAs encoded by APMV was only 1.8%, even at the late infection stage (8 h.p.i.) (Fig. 3a), suggesting that the contribution of viral tRNA to the cellular tRNA pool may be limited. In addition, the codons decoded by the APMV-encoded tRNAs did not show significantly different ribosome occupancy (that is, elongation speed) compared to other codons in viral mRNAs (Extended Data Fig. 6a).

To investigate whether the virus infection altered the tRNA pool composition to favour viral mRNA translation, we calculated the proportion of each tRNA type in the tRNA pool (including amoeba nuclear and APMV tRNAs). We used $W$-score to quantify the tRNA supply level. For a given codon type, $W$-score was calculated by summing the

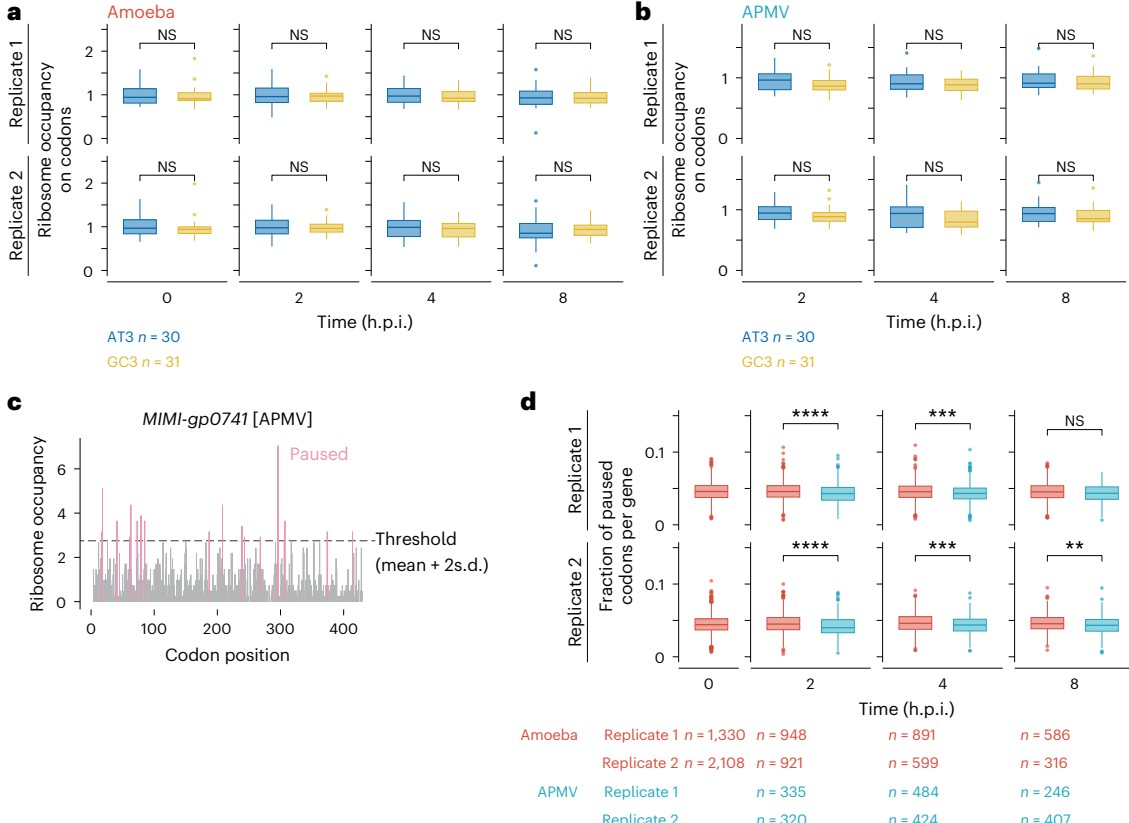

**Fig. 2 | Smooth ribosome traversal on APMV mRNAs. a,b**, Box plots of ribosome occupancy on AU3 and GC3 codons in amoeba (**a**) and APMV (**b**) mRNAs at different infection stages. AU3, codons ending with A or U; GC3, codons ending with G or C. **c**, Distribution of ribosome footprints along the open reading frames of *MIMI-gp0741* (locus tag of an APMV gene) in the APMV genome for replicate 1 at 2 h.p.i. The A-site position of the ribosomes is shown. The first four codons and stop codons were not included and are not plotted. The grey horizontal line (Threshold) indicates the cut-off of pausing (mean + two standard deviations (2 s.d.)). Pause sites are shown in pink. **d**, Box plots of the fraction of paused codons per gene in the amoeba and APMV genomes at different infection stages. Box plots as in Fig. 1. The *P* values were calculated using one-sided Wilcoxon rank sum test. *$P < 0.05$; **$P < 0.01$; ***$P < 0.001$; ****$P < 0.0001$.

proportions of cognate and wobble-paired tRNAs (see Methods)[30]. Essentially, *W*-scores were stable during the APMV infection; high correlation of the *W*-scores was observed between time points (Fig. 3b). In naive amoeba (that is, 0 h.p.i.), the *W*-scores were higher for GC3 than they were for AU3 codons (Fig. 3c and Extended Data Table 1). This finding corresponds with codon usage in the amoeba genome, which is biased to GC3 codons (Extended Data Fig. 1) and probably contributed to the smooth elongation of GC3 codons (Fig. 2a) with no shortage of tRNA supply. The high tRNA supply for GC3 codons was maintained throughout the infection, indicating that viral infection did not largely change the composition of the global tRNA pool.

To quantitatively assess the imbalance of tRNA supply and demand during APMV infection, we calculated the balance score, which borrowed the idea of the normalized translation efficiency index[31]. In brief, we calculated the ratio between tRNA supply and codon usage, considering the mRNA abundance (see Methods). In naive amoeba, the AU3 codons generally had a higher balance score than the GC3 codons (Fig. 3d), indicating that tRNAs for AU3 codons are 'excessive' despite their concentration being lower than that for GC3 codons. During APMV infection, the high expression of viral mRNAs led to a decrease in the balance score for AU3 codons, whereas the balance score for GC3 increased (Fig. 3d and Extended Data Table 2). Together, these findings indicate that the tRNA supply and demands inside the cell seemed unsuitable for the translation of AU3-rich viral mRNAs (Fig. 3), even though protein synthesis from viral mRNAs was rather smooth (Fig. 2d).

## Distinct ribosome elongation environment for viral mRNAs

The conflict between tRNA supply and demand for viral mRNA translation is based on our assumption that cellular resources, such as tRNAs, are equally available for host and viral mRNAs. Thus, our results suggest that the translation environment for viral mRNAs was not the same as that for host mRNAs. Consistent with this scenario, we observed differences in tRNA accessibility in host and viral mRNAs. Given that long (29–30 nt) and short (21–22 nt) footprints represent ribosomes with or without tRNA at the A-site[29], we defined the short footprint ratio as the ratio of short footprints to total footprints for a given codon type at the A-site to represent tRNA accessibility. The short footprint ratio was higher (that is, lower tRNA accessibility) on AU3 codons than it was on GC3 codons in amoeba mRNAs, irrespective of the viral infection stage (Fig. 4a). This finding is consistent with the lower tRNA supply for AU3 codons compared with the tRNA supply for GC3 codons (Fig. 3c). There was no such difference in the short footprint ratio for viral mRNAs (Fig. 4b). The same results were obtained when the ratio of short footprint over long footprints was considered (Extended Data Fig. 7a,b). The correlation between viral codon occupancy and host codon occupancy decreased at the late infection stage (8 h.p.i.) (Extended Data Fig. 6a), indicating divergence in codon-specific elongation speed between viral and host mRNAs. These data suggest that the translation environment of viral mRNAs differed from that of amoeba mRNAs.

To further investigate the cytosolic translation situation during viral infection, we set out a reporter assay to monitor the effects of AT-rich and GC-rich sequences in amoeba with or without APMV infection (Extended Data Fig. 8a). We used an established reporter system,

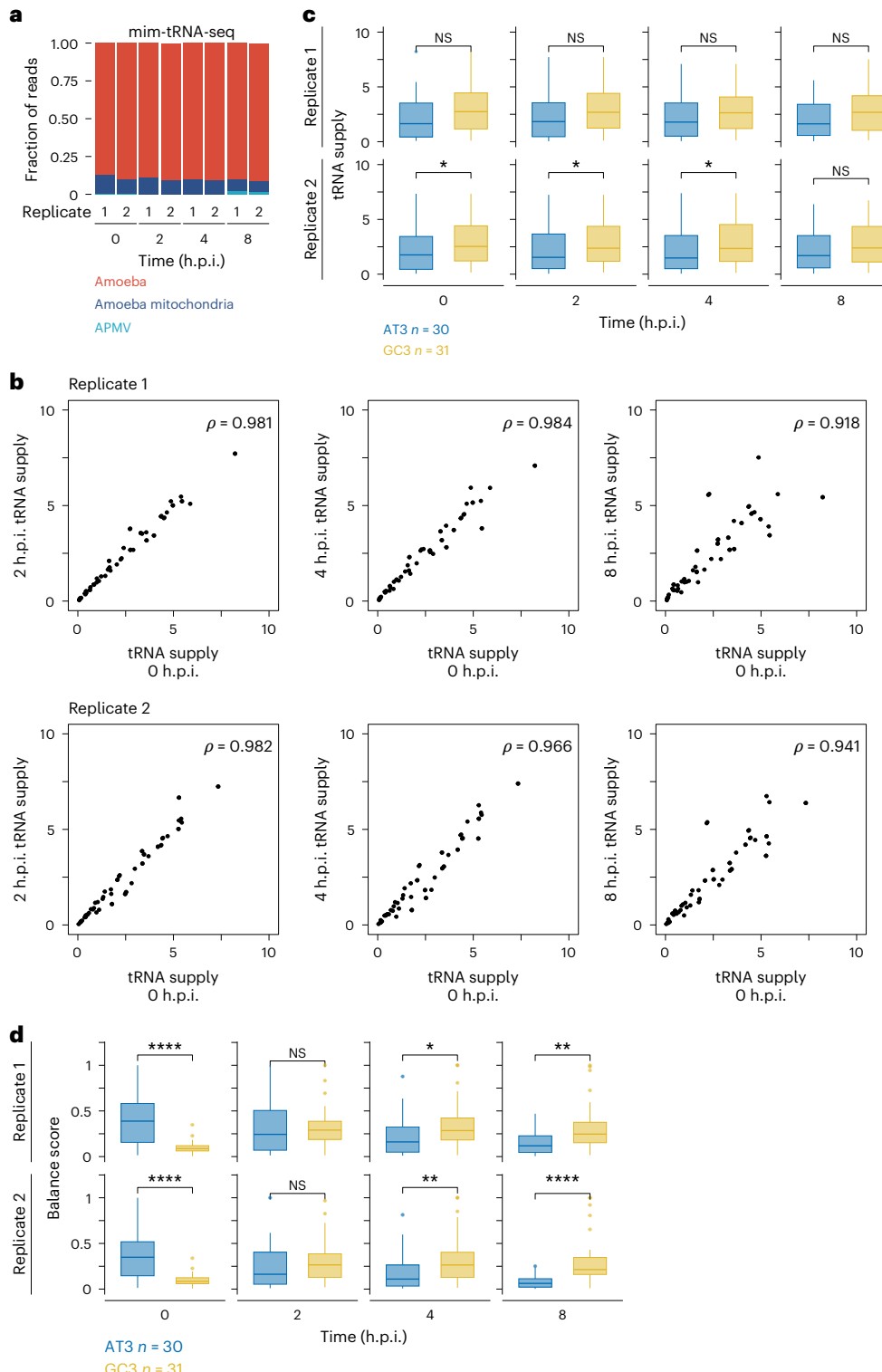

**Fig. 3 | Stable tRNA pool during infection conflicts with translation elongation on APMV mRNAs. a**, Fraction of reads that aligned to the indicated genomes in the mim-tRNA-seq analysis at different infection stages. **b**, Correlation of tRNA supply evaluated by $W$-score among the different infection stages. $\rho$, Spearman's correlation coefficient (two-tailed). **c**, Box plots of $W$-scores for AU3 and GC3 codons at different infection stages. **d**, Box plots of balance scores for tRNA for AU3 and GC3 codons at different infection stages. Box plots as in Fig. 1. The $P$ values were calculated using one-sided Wilcoxon rank sum test. *$P < 0.05$; **$P < 0.01$; ***$P < 0.001$; ****$P < 0.0001$.

which is sensitive to translation elongation effects[32–35]. This reporter consists of an N-terminal *Renilla* luciferase (Rluc) and a C-terminal firefly luciferase (Fluc) in frame, with the open reading frame (ORF) region of the test in the middle. Viral 2A sequences, which induce self-cleavage of peptide bonds during translation[36], were inserted in the middle (Extended Data Fig. 8a). The ratio of the two individual luciferase proteins allows measurement of protein synthesis rate in the region between luciferases. Here, we investigated the effects of AT-rich

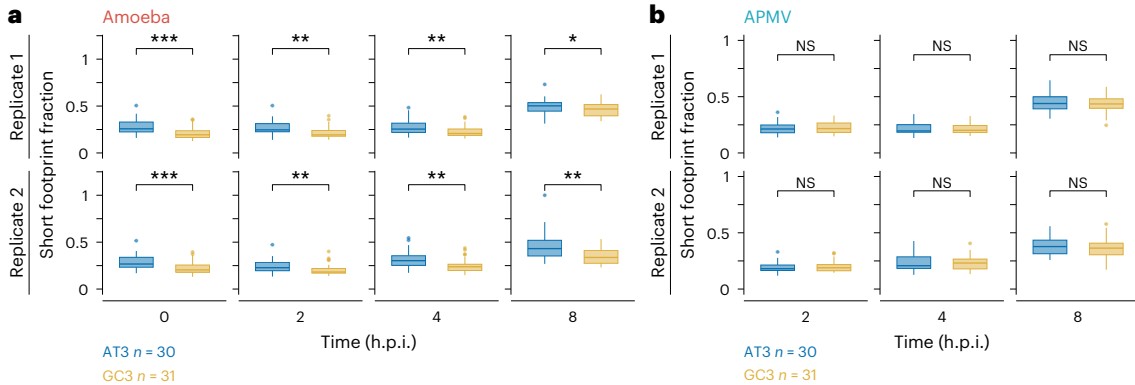

**Fig. 4 | Difference in tRNA accessibility between amoeba and APMV mRNAs. a,b,** Box plots of the short footprint fraction on AU3 and GC3 codons in amoeba (**a**) and APMV (**b**) mRNAs at different infection stages. Box plots as in Fig. 1. The *P* values were calculated using one-sided Wilcoxon rank sum test. *P < 0.05; **P < 0.01; ***P < 0.001; ****P < 0.0001.

ORF originating from TATA-box binding protein (TBP) gene of APMV and GC-rich ORF from actin gene of amoeba (Extended Data Fig. 8a).

Consistent with genomic sequences, the GC-rich region of the reporter resulted in a higher Fluc/Rluc ratio than the AT-rich region, suggesting a smoother elongation for GC-rich mRNAs (Extended Data Fig. 8b). The slow elongation of the AT-rich region was not improved, while a smoother elongation of the GC-rich region was maintained upon viral infection (Extended Data Fig. 8b). These observations suggest that the cytosolic environment performs a smoother translation of GC-rich mRNAs, further supporting the requirement of a unique environment for viral protein synthesis.

### Viral mRNAs are locally translated

As ribosomes were found to localize around the viral factory[17,18], we hypothesized that viral mRNA translation may occur in a distinct subcellular location near the viral factory. To investigate this hypothesis, we combined 4′,6-diamidino-2-phenylindole (DAPI) staining, FISH and FUNCAT[16,21] to determine the location of the viral factory, viral mRNAs, amoeba ribosomes and newly synthesized proteins. The FUNCAT signal was enriched around the periphery of the viral factory (Fig. 5a). Similarly, FISH for amoeba rRNA showed that a sub-population of ribosomes was also enriched in this region (Fig. 5a), which is consistent with previous tomography results[18], suggesting that protein synthesis occurred at the viral factory periphery. Moreover, this local translation milieu contained the APMV-encoded *mcp* mRNAs, probed by single-molecule mRNA fluorescence in situ hybridization (sm-mRNA FISH) (Fig. 5b). Thus, our data suggest that APMV created a unique environment tailored for viral mRNA translation at the periphery of the viral factory.

### Discussion

A consonant pattern between codon usage and tRNA concentration was thought to be important for translation[2,3]. However, APMV has a codon usage pattern that deviates from its host, suggesting potential problems in viral gene translation. The deep sequencing-based techniques (RNA-seq, Ribo-seq and mim-tRNA-seq) showed that APMV genes were smoothly translated without reduced accessibility to tRNAs for frequently used codons. The cell biology methods (FUNCAT and FISH) showed that viral mRNAs, host rRNA and protein synthesis were co-localized at the periphery of the viral factory. These findings suggest that APMV mitigates the codon usage conflict by creating a local translation milieu for its mRNAs.

Viral mRNAs dominated the mRNA pool at the late stages of infection, which is consistent with a previous study[8]. However, this increase was not perfectly reflected in the footprint proportion (Fig. 1b,c), leading to a decline in translation efficiency of viral mRNAs (Fig. 1h).

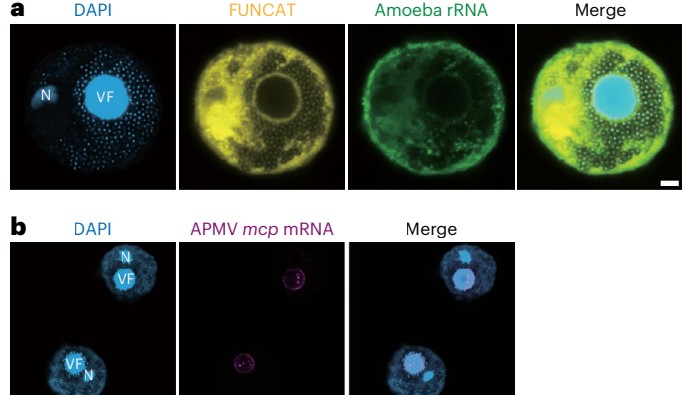

**Fig. 5 | Local translation at the peripheral region of the viral factory during APMV infection. a,** Microscopy images for DAPI (light blue), FUNCAT (yellow) and rRNA FISH (green) of APMV-infected amoeba cells at 12 h.p.i. **b,** Microscopy images for DAPI (light blue) and sm-mRNA FISH for APMV *mcp* mRNA of APMV-infected amoeba cells at 18 h.p.i. N, amoeba nucleus; VF, viral factory. Scale bars, 3 μm (**a**) and 8 μm (**b**).

A low translation efficiency during viral infection has been reported for influenza A virus and SARS-CoV-2 (refs. 37,38). It suggests that overwhelming the host's translation machinery through massive mRNA synthesis might be a common strategy shared by diverse viral taxa. Consistently, we found a general increase of A-site empty ribosomes, represented by short footprints along the viral infection stages (Extended Data Fig. 3f,g), probably because of the global tRNA or amino acid shortage.

In terms of translation dynamics and tRNA pool, we found that APMV genes were smoothly translated without significantly altering the tRNA composition in the host cell (Figs. 2d and 3). This unaltered cellular tRNA pool aligns with the previous report on vaccinia virus and influenza A virus[39]. One possible explanation is that AU-rich mRNAs form fewer secondary structures than GC-rich mRNAs, facilitating a smooth translation. However, this cannot explain the observed higher tRNA accessibility for AU3-codons on viral mRNAs than that on host mRNAs (Fig. 4). These observations suggest that a local tRNA pool could be selectively used for viral mRNAs. APMV may achieve smooth translation by generating a local translation environment within the cell.

In terms of the subcellular organization within the infected cells, the viral factory is a unique intracellular structure that is formed by many large DNA viruses during infection. For APMV, DNA replication

and transcription occur in the viral factory[15,39], whereas endoplasmic reticulum, ribosomes and the viral translation initiation factor SUI1 have been observed outside the viral factory[15,17,18]. In this study, we showed that viral mRNAs surrounded the viral factory (Fig. 5b), and protein synthesis occurred in the same region (Fig. 5a). Other translation-related proteins and tRNAs may also be recruited to the same place, which would further increase the local concentration of translation machineries. Other viruses might also use similar translation strategies. For example, poxviruses, asfarviruses and iridoviruses have been reported to induce changes in cytoskeleton[40,41], recruitment and local enrichment of proteins[42] and even re-localization of organelles such as mitochondria[40,43]. Moreover, endoplasmic reticulum membrane recruitment and protein re-localization have been observed for RNA viruses[44–46], indicating that subcellular remodelling is a common feature of viral infections across diverse virus families.

APMV likely uses a local translation environment to overcome the codon–tRNA mismatch. However, the reason for the mismatch (that is, AT-rich genomic feature) still needs an explanation. One hypothesis is that the AT-rich genome arises from mutational biases due to the lack of DNA repair genes, as proposed for many AT-rich viruses and intracellular microorganisms[47,48]. However, this scenario might not apply to APMV, which has a large set of DNA repair genes[49]. Instead, our results brought another possibility for the origin of the genomic feature. The AT-rich genome might be selectively advantageous by allowing access to the excessive tRNAs for AU3 codons at the early stage of infection (Fig. 3d). This possibility may apply to other viruses, as APMV is not the only virus that has a deviated codon usage with its host[50–52]. In the case of APMV, this advantage is likely to cease at later infection stages with high viral mRNA load (Fig. 3d). The viral mRNA load at later infection stages appears to cause a translation-associated bottleneck, which may be alleviated through the acquisition of translation-related genes by APMV and related viruses to build favourable translation environments. Viruses in the order *Imitervirales* are atypical among viruses by encoding abundant tRNA genes and translation-related proteins such as aminoacyl-tRNA synthetases[53,54]. It was also noted that viral aminoacyl-tRNA synthetases were found to be frequently associated with AT-rich codons[53]. Similar hypotheses have been proposed for the incorporation of tRNA genes into other viral genomes[55–57].

In this study, we proposed that the creation of a subcellular translation environment is a key strategy for APMV to overcome the discrepancy between host and viral codon usages. Virus-induced subcellular structures have been observed across a wide range of viruses, from prokaryotic to eukaryotic and from RNA to DNA viruses[17,58,59]. Although the spatial association between these viral subcellular structures and translation is largely unknown for other viruses, they may also form a local environment for viral mRNA translation.

## Methods

### Cells and viruses
*A. castellanii*, strain Neff (American Type Culture Collection (ATCC), 30010) was cultured in peptone–yeast–glucose (PYG) medium at 28 °C. Although a recent study proposed a reclassification of this strain[60], it has not yet been recognized by the National Center for Biotechnology Information (NCBI). We used the original naming for consistency with the NCBI database. APMV was used as a prototype of mimiviruses[61]. The virus titre was determined by the 50% tissue culture infectious dose (TCID$_{50}$) assay using *A. castellanii* cells.

### Lysate preparation
Lysate for Ribo-seq, RNA-seq and mim-tRNA-seq was prepared as described previously[28]. In brief, *A. castellanii* cells were seeded on 10 cm plates at $3 \times 10^6$ cells per ml supplemented with 10 ml of PYG medium. Amoeba cells were infected with APMV at a multiplicity of infection of 10. The plates were incubated for 1 h at room temperature to allow viral absorption, then the medium was replaced with fresh PYG medium. The medium replacement timing was set to 0 h.p.i., and APMV-infected cells were collected at 2, 4 and 8 h.p.i. For mock treatment, PYG medium was used and collected at 0 h.p.i. Two independent replicates were prepared for each time point.

Cells were treated with 100 µg ml$^{-1}$ cycloheximide for 1 min, then scraped down and pelleted at 300$g$ at 4 °C for 3 min, washed with 1 ml of PBS containing 100 µg of cycloheximide (Sigma–Aldrich) and 100 µg of chloramphenicol (Wako Pure Chemical Industries) and pelleted again at 300$g$ at 4 °C for 2 min. The cell pellets were resuspended with 600 µl lysis buffer (20 mM Tris pH 7.5, 150 mM NaCl, 5 mM MgCl$_2$, 1 mM dithiothreitol, 1% Triton X-100, 100 µg ml$^{-1}$ cycloheximide and 100 µg ml$^{-1}$ chloramphenicol), treated with 5 µl of TURBO DNase (2 U µl$^{-1}$, Thermo Fisher Scientific) on ice for 10 min, then centrifuged at 20,000$g$ at 4 °C for 10 min. The supernatant was stored at −80 °C before library construction. RNA concentration of cell lysates was measured with a Qubit RNA BR Assay Kit (Thermo Fisher Scientific).

### Preparation of Ribo-seq library
Ribo-seq was performed as described previously[28]. The cell lysate containing 10 µg total RNA was incubated with 2 U of RNase I (LGC Biosearch Technologies) in a 300 µl reaction mixture at 25 °C for 45 min. The RNase digestion was stopped by adding 10 µl of SUPERase•In (Thermo Fisher Scientific). Ribosomes were isolated by sucrose cushion ultracentrifuged at 100,000 r.p.m. (543,000$g$) for 1 h at 4 °C with a TLA110 rotor and an Optima MAX-TL ultracentrifuge (Beckman Coulter). Subsequently, RNA was purified with TRIzol-LS reagent (Thermo Fisher Scientific) and a Direct-zol RNA Microprep kit (Zymo Research). RNA fragments of 17–35 nt were gel-purified after running polyacrylamide gel electrophoresis. The isolated RNA fragments were ligated to custom-made pre-adenylated linkers containing unique molecular identifiers and barcodes for library pooling using T4 RNA ligase 2, truncated KQ (New England Biolabs). rRNA was depleted using a Ribo-Zero Gold (Human/Mouse/Rat) kit (Illumina), followed by a pull-down using RNAClean XP beads (Beckman Coulter). The rRNA-depleted samples were reverse transcribed with ProtoScript II (New England Biolabs), circularized with CircLigase II (LGC Biosearch Technologies) and PCR-amplified using Phusion polymerase (New England Biolabs). The libraries were sequenced on an Illumina HiSeqX system (Illumina) with the 150 nt paired-end read option (Supplementary Table 1).

### Preparation of RNA-seq library
Total RNA was purified from the cell lysate using TRIzol-LS and a Direct-zol RNA Microprep kit (Zymo Research). RNA-seq libraries were prepared with TruSeq Stranded Total RNA Library Prep Gold (Illumina). The library was sequenced on a HiSeqX system (Illumina) with the 150 nt paired-end reads option (Supplementary Table 1).

### Preparation of mim-tRNA-seq library
The tRNA fraction was collected from the lysate using a mirVana miRNA Isolation Kit (Thermo Fisher Scientific) and deacylated by incubating at 37 °C for 45 min in 100 mM Tris–HCl pH 9.0. The library preparation scheme was adapted from the method for Ribo-seq[28], using 80 ng of tRNAs. Then, 2 µl of 1.25 µM reverse transcription primer (NI-802)[28] was hybridized to linker-ligated tRNA dissolved in 10 µl of RNase-free water by denaturing at 82 °C for 2 min and cooling at 25 °C for 5 min in a thermocycler. Reverse transcription was conducted in 21.6 mM Tris–HCl pH 7.5, 5.4 mM MgCl2 and 486 mM KCl containing 5.4 mM dithiothreitol, 0.54 U µl$^{-1}$ SUPERase•In (Thermo Fisher Scientific) and 540 nM TGIRT-III (InGex) in an 18.5 µl reaction mixture and pre-incubated at 42 °C for 10 min in a thermocycler. Next, 2.5 µl of 10 mM dNTP was added to the tube to make a 20 µl reaction mixture and incubated at 49 °C for 16 h in a thermocycler. Subsequent circularization and PCR amplification were conducted as in the Ribo-seq[28]. The library was sequenced on a HiSeqX system (Illumina) with the 150 nt paired-end reads option (Supplementary Table 1).

## Construction of a non-coding RNA database

We constructed a non-coding RNA (ncRNA) database by collecting annotated sequences from databases and de novo prediction. The annotated ncRNA sequences of *A. castellanii* were downloaded from the RNAcentral database (17 August 2023) and searched by queries '*Acanthamoeba castellanii*', which hit to the following two TAXONOMY levels, 'TAXONOMY 5755' and 'TAXONOMY 1257118' (RNAcentral Consortium, https://rnacentral.org)[62]. The TAXONOMY 5755 search identified 241 ncRNA records: 205 rRNAs, 32 small ncRNAs, 3 RNase P RNAs, 1 RNase MRP RNA and 1 group I intron. The TAXONOMY 1257118 search identified 205 ncRNA records: 44 rRNAs, 149 small non-coding RNAs, 1 RNase P RNA, 1 RNase MRP RNA, 4 SRP RNAs and 2 vault RNAs.

For de novo prediction, the nuclear genome sequence of *A. castellanii* was downloaded from the latest chromosome-scale genome assembly[14]. The mitochondrial (NC_001637.1) and APMV (NC_014649.1) genome sequences were downloaded from the RefSeq database and used for de novo ncRNA prediction. rRNA genes were predicted using RNAmmer (v.1.2)[63], and tRNA genes were predicted using tRNAscan-SE (v.2.0.12; parameters: -HQ for amoeba, -G -HQ for APMV, -mt -Q for mitochondria)[64].

## Read mapping

For the Ribo-seq and RNA-seq reads, data were processed as described previously[65]. In brief, adapter trimming was performed with fastp (v.0.21.0)[66]. Reads were mapped to the ncRNA sequences using STAR (v.2.7.0a)[67] to remove reads that originated from rRNA genes and other ncRNA genes in silico. The remaining reads were mapped to the merged genomes of the virus, amoeba nucleus and mitochondria, using STAR (v.2.7.0a). BAM file indexing and read extraction were performed using SAMtools (v.1.10)[68]. Offsets for footprints of different lengths were determined by checking the enrichment of footprints around the start and stop codons of genes (Supplementary Table 3). The quality of ribosomal footprints was checked by mapping status around start and stop codons (Extended Data Figs. 2 and 3a).

The mim-tRNA-seq reads were quality controlled using fastqc (v.0.12.1; http://www.bioinformatics.babraham.ac.uk/projects/fastqc/), and regions after 97 nt (quality ≤ 20) were trimmed using fastx-trimmer (v.0.0.14; http://hannonlab.cshl.edu/fastx_toolkit/). The trimmed reads were mapped to the ncRNA sequences (except for tRNAs) using Bowtie2 to remove other ncRNAs[69]. The unmapped reads were extracted and mapped to the tRNA sequences using Bowtie2.

## Gene filtering, quantification and normalization

To quantify gene expression levels, reads per kilobase per million mapped reads (RPKM) values were calculated for protein-coding genes in the amoeba and APMV genomes based on the total number of reads that uniquely aligned to the coding regions of the amoeba or virus, respectively.

Footprints that mapped to the first five codons (including the start codon) or the last five codons (including the stop codon) were not counted. Open reading frames with total RNA and footprint RPKM values < 10 in any replicate were discarded. After filtering, 10,575 and 955 protein-coding genes from the amoeba nuclear genome and APMV genome, respectively, remained for subsequent analyses.

Transcripts per million kilobase (TPM) values were used to compare host and viral gene expression levels. To analyse the gene expression patterns in the host or virus, TPM values were recalculated separately for the host and virus genes. Translation efficiency was calculated using DEseq2 (ref. 70) as a mean fold-change between footprint and read counts, considering replicates.

## Gene clustering and enrichment analysis

After filtering lowly expressed genes, *Z*-score normalization was performed on each gene in one replicate. The optimal number of *K*-means clusters was determined based on TPM values using the R package

factoextra (v.1.0.7; https://rpkgs.datanovia.com/factoextra/index.html). KEGG pathway and KEGG orthologue groups were assigned using the eggNOG-mapper webserver (v.2.1.1.2)[71] (http://eggnog-mapper.embl.de/) with the following settings: *e*-value < 1 × 10⁻⁵, bit score ≥60, identity ≥60%, query coverage ≥20% and subject coverage ≥20%. The KEGG Pathway database for amoeba was built with the assigned KEGG Orthology terms using a homemade R code. Enrichment analyses were performed for the top 2,000 genes with the highest standard deviation using clusterProfiler (v.4.8.3)[72] (Supplementary Table 2).

## Quantification of tRNA expression

For tRNA read mapping, reads that met the following criteria were retained: (1) length ≥20 bp; (2) uniquely mapped to tRNA genes that encode the same anticodons; (3) anticodon of mapped tRNA was not 'NNN', 'TTA' (that is, tRNA suppressor) or 'TCA' (that is, tRNA for selenocysteine); and (4) reads mapped to the same genome. These criteria ensured that reads with unique mapping and reads that mapped to multiple genes but encoded the same type of anticodon were kept for quantification.

The composition of the cellular tRNA pool was calculated as the proportion of tRNA species in each sample. Because tRNA and a codon can pair by wobble pairing, we used absolute adaptiveness (*W*) to measure the tRNA supply to each type of codon[30]. The *W*-score of a codon was computed as the weighted sum of the selection constraint ($s_{ij}$) and tRNA ratio (tRNA$_j$), considering both Crick's rules and wobble pairing. The selective constraint ($s_{ij}$) represents the difficulty of the pairing between codon *i* and anticodon *j*.

$$W_i = \sum_{j=1}^{n_i}(1-s_{ij})\text{tRNA}_j \tag{1}$$

In brief, to infer $s_{ij}$, we performed Nelder–Mead optimization on Spearman's correlation between the expression level and computed tRNA adaptation index (tAI) of corresponding genes using the Python package scipy[73]. The tAI is a measure of the tRNA usage by coding sequences[30]. For a gene *g*, tAI$_g$ was computed as the geometric mean of the relative adaptiveness $w_i$ of its codons. If tAI$_g$ = 1, it indicates the gene *g* is well supported by the tRNA pool, whereas tAI$_g$ close to 0 indicates that the gene is poorly supported.

$$\text{tAI}_g = \left(\prod_{k=1}^{l_g} w_{i_{kg}}\right)^{\frac{1}{l_g}} \tag{2}$$

Here, $i_{kg}$ is the codon at the *k*th position in the gene *g*, and $l_g$ is the length of the gene in codons (excluding the stop codon). Consequently, tAI$_g$ estimates the amount of adaptation of a gene *g* to its genomic tRNA pool.

In equation (2), the relative adaptiveness $w_i$ is the normalized absolute adaptiveness (*W*-score). The relative adaptiveness $w_i$ was calculated as the ratio between the $W_i$ for each codon and the maximum *W*. Thus, $w_i$ = 1 indicates that the codon *i* is the most adapted codon to the tRNA pool, and $w_i$ close to 0 indicates that the codon is very poorly adapted.

Originally, the maximum *W* was set as the highest *W* among all codons. To avoid making this value biased to being much higher than other values, we used the geometric mean of the three highest *W* values ($W_N$) as the maximum *W* value for computation. $W_N$ was used to calculate the relative adaptiveness of each codon.

$$W_N = \sqrt[3]{\prod_{i\in\text{top3}} W_i} \tag{3}$$

$$w_i = \min\left(1, \frac{W_i}{W_N}\right) \tag{4}$$

After calculating tAI for each gene, we performed parameter optimization for $s_{ij}$ using the top 2,000, 3,000, 4,000 and 5,000 amoeba genes based on the ranking of the footprint TPM separately and initialized the parameter set from [0, 0, 0, 0], [0.25, 0.25, 0.25, 0.25, 0.25], [0.5, 0.5, 0.5, 0.5, 0.5], [0.75, 0.75, 0.75, 0.75, 0.75], [1, 1, 1, 1]. The $s_{ij}$ set was [1, 0, 1, 0.0065956, 0.90832489] ($\rho \approx 0.450$), which represents the wobble pairing between A:I, C:I, G:U, U:G and A:G (for the codon–ATA:tRNA–GAT pair), respectively.

## Codon usage and balance score

For codon usage, $U_i$, the total number of occurrences of codon type $i$ was computed by summing up the occurrences of codon $i$ in the gene $g$, denoted as $c_{ig}$, weighted by the mRNA abundance $TPM_{RNA,g}$.

$$U_i = \sum_{g=1}^{G} TPM_{RNA,g} \times c_{ig} \qquad (5)$$

We defined codon usage $cu_i$ as a relative estimate of how often each codon is used. We computed $cu_i$ by rescaling $U_i$ to have a maximum value of 1. $U_{max}$ in equation (7) is the geometric mean of $U_i$ of the top three codons.

$$cu_i = \min\left(1, \frac{U_i}{U_{max}}\right) \qquad (6)$$

$$U_{max} = \sqrt[3]{\prod_{i \in top3} U_i} \qquad (7)$$

The balance score is defined as the ratio of tRNA availability, the relative adaptiveness of each codon ($w_i$), and codon usage ($cu_i$), then linearly rescaled to have a maximum value of 1 (ref. 31).

$$ratio_i' = \frac{w_i}{cu_i} \qquad (8)$$

$$ratio_{max}' = \sqrt[3]{\prod_{i \in top3} ratio_i'} \qquad (9)$$

$$Balance\ score_i = \min\left(1, \frac{ratio_i'}{ratio_{max}'}\right) \qquad (10)$$

A codon with ratio′ > 1 was defined as 'excessive tRNAs', indicating that the tRNA supply was higher than the codon usage, and ratio′ < 1 was defined as 'moderate tRNAs', indicating the tRNA supply was lower than the codon usage. The balance score is derived from this score by rescaling the range from 0 to 1.

## Codon occupancy

The relative ribosome density at the $k$th codon of gene $g$ ($e_{kg}$) was defined as the number of ribosome footprints at position $k$ ($RPF_{kg}$) to the average number of ribosome footprints per codon of gene $g$ ($R\bar{P}F_g$).

$$e_{kg} = \frac{RPF_{kg}}{R\bar{P}F_g} \qquad (11)$$

$$R\bar{P}F_g = \frac{RPF_g}{l_g} \qquad (12)$$

The ribosome occupancy of a given type of codon $i$ was the mean value of all positions where the corresponding codon is $i$. Only genes with $R\bar{P}F_g > 1$ were included in this analysis. The first four codons in each gene were excluded when calculating codon occupancy[74]. Stop codons were also not included in the analysis.

## Short footprint ratio

The short footprint ratio of codon type $i$ was defined as the ratio between the number of short footprints ($RPF_s$) mapped to a codon $k$ and the total number of footprints ($RPF_T$) mapped to codon type $i$. Genes that were used in the codon occupancy analysis were used in this study. The start and stop codons were excluded from the calculation. Short footprint ratio was repeated using 21–22 nt and 29–30 nt footprints alone to examine the signal (Extended Data Fig. 7a,b).

$$Short\ footprint\ ratio\ for\ codon\ type\ i = \frac{RPF_{S,i}}{RPF_{T,i}} \qquad (13)$$

## Identification of a putative pausing site

For each gene, we calculated the mean and standard deviation of the relative ribosome density as $\bar{e}_g$ and $sd_g$. We considered positions as putative pausing sites when $e_{kg}$ exceeds the mean ($\bar{e}_g$) plus two standard deviations ($s_{dg}$) of the gene[34]. The number of putative paused sites was summed based on their codon type and the original gene. These numbers were normalized by the total occurrences of codon types and gene length, excluding the first four codons and the stop codon.

## FUNCAT, sm-mRNA FISH, rRNA FISH and DAPI staining

*A. castellanii*, strain Neff (ATCC 30010) cells were seeded in PYG medium at $10^5$ cells per ml in a 24-well plate (Thermo Fisher Scientific) with a final volume of 1 ml per well. The cells were incubated to attach to the dish bottom at 25 °C for at least 30 min. Subsequently, the cells were infected with APMV at a multiplicity of infection of 1. The plates were centrifuged for 30 min at 1,000g at room temperature to synchronize the infection. Then, the medium was exchanged for fresh PYG medium to minimize the possibility of re-infections.

For each well, the 1 ml cell solution was collected and transferred into a 1.5 ml tube (Eppendorf) and centrifuged at 5,000g for 5 min at room temperature. After removing 80% of the supernatant, the remaining volume was resuspended using a vortex. Then, 50 µl of this concentrated cell suspension was added to each well of a microscopy slide (Marienfeld). The cells were incubated for 1 h to attach to the microscopy slide. The supernatant was removed, and the cells were fixed using 4% paraformaldehyde for 10 min at room temperature. The cells were washed using Milli-Q water and permeabilized with 70% ethanol for 2 h at 4 °C.

FUNCAT, sm-mRNA FISH and rRNA FISH were performed as described previously[16]. FUNCAT was developed from bio-orthogonal noncanonical amino acid tagging[21,75–78]. In brief, homopropargylglycine was added at a concentration of 50 µM at least 30 min before sampling. To identify newly synthesized proteins, an azide-bearing affinity tag was covalently attached by 'click chemistry'. For this, 221 µl Page's amoeba saline buffer (PAS, ATCC medium 1323) was mixed with 12.5 µl of 100 mM sodium ascorbate and 12.5 µl of 100 mM aminoguanidine hydrochloride. Separately, 1.25 µl of 20 µM CuSO₄, 1.25 µl of 100 µM Tris[(1-hydroxypropyl-1H-1,2,3-triazol-4-yl)methyl]amine (THPTA) and 0.3 µl of 5 mM Cy3-azide dye were mixed and left to react in the dark for 3 min at room temperature. The two solutions were mixed carefully, and 10 µl was added to each well of the microscope slides. The slides were incubated in the dark for 30 min at room temperature and then washed with PAS.

For the sm-mRNA FISH, 35 20 nt probes were designed to target the *mcp* gene of APMV[16] using the Stellaris RNA FISH Probe Designer of LGC Biosearch Technologies (https://www.biosearchtech.com/support/tools/design-software/stellaris-probe-designer). To each well on the microscopy slides, 10 µl of hybridization buffer (2× SCC, 100 mg ml⁻¹ dextran sulfate, 2 mM ribonucleoside-vanadyl complex, 0.2 mg ml⁻¹ BSA, 1 mg ml⁻¹ *E. coli* tRNA and 20% formamide) containing the probe mix (with each probe labelled with the fluorophore Cy5) was added at a working probe solution of 437.5 nM. The slides were placed

in a moist chamber and incubated at 37 °C for 20 h. The hybridization buffer was washed away with a wash buffer (2× SCC in nuclease-free water) at 39 °C for 10 min. Then, the slides were dipped in ice-cold water and air dried.

For the rRNA FISH, 10 μl of hybridization buffer (0.9 M NaCl, 20 mM Tris–HCl pH 8.0, 0.01% SDS and 25% formamide) and 1 μl of Euk516 probe (5′-GGAGGGCAAGTCTGGT-3′; labelled with the fluorophore Fluos) were added to each well of the microscopy slides. Hybridization was performed in moist chambers at 46 °C for 1.5 h. Subsequently, slides were washed with a wash buffer (20 mM Tris–HCl pH 8.0, 5 mM EDTA and 0.149 M NaCl) and kept in the wash buffer in a 50 ml tube at 48 °C for 10 min. Then, the slides were dipped in ice-cold Milli-Q water and air dried.

After the sm-mRNA FISH or rRNA FISH, DAPI (1 μg ml$^{-1}$) was added to the wells on the slides, incubated for 3 min and washed off with nuclease-free water and air dried. The slides were observed using a confocal laser-scanning microscope (Leica SP8).

### Plasmid construction

The plasmid for exogenous expression (pAcasTBP-GFP) in amoeba cells was constructed as follows. A promoter sequence of the *A. castellanii* TBP gene, a fused ORF of the enhanced green fluorescence gene (GFP) and the neomycin resistance gene (NEO), and the SV40 early polyA sequence were PCR-amplified from *A. castellanii* genomic DNA, a synthetic plasmid (VectorBuilder), and pIZ/GFP[79], respectively, using KOD One PCR Master Mix (Dye-free 2×PCR Master Mix) (TOYOBO). These fragments were cloned into the pBR322 backbone, which was PCR-amplified from pUC19, using the In-Fusion HD Cloning Kit (TaKaRa). Subsequently, the SV40 late polyA sequence and an additional TBP promoter sequence were PCR amplified from the synthetic plasmid and the *A. castellanii* genomic DNA, respectively, and introduced between the GFP and NEO ORFs, using the In-Fusion HD Cloning Kit.

To generate pAcasTBP-Rluc-2A-GC-2A-Fluc and pAcasTBP-Rluc-2A-AT-2A-Fluc, fragments containing *Renilla* luciferase (RLuc) or firefly luciferase (FLuc) with self-cleavage 2 A sequences were PCR-amplified from plasmid psiCHECK2-2A-3xFLAG-SBP-2A[34]. The GC-rich sequence from the amoeba actin gene (BAESF_03529.mRNA.1)[14] and the AT-rich sequence from the APMV TBP ORF (RefSeq GeneID: 9925078, protein ID: YP_003986960.1) were PCR-amplified from *A. castellanii* and APMV genomic DNA, respectively. The amplified DNA was cloned into pAcasTBP-GFP that was linearized by PCR using the In-Fusion Snap Assembly Master Mix (TaKaRa).

APMV genomic DNA was prepared as described previously[80]. *A. castellanii* genomic DNA was prepared using NucleoSpin Tissue XS (TaKaRa).

### Reporter assay

Amoeba cells at the growth phase were collected, and $5 \times 10^5$ cells were seeded in a 35 cm Nunc EasYDish (Thermo Fisher Scientific) with 2 ml of PYG medium. Then 5 μg of plasmids was transfected with 10 μl of Poly-Fect Transfection Reagent (Qiagen) in 100 μl of phosphate-buffered saline and 600 μl of PYG. One day after transfection, 1 ml of floating cells was transferred into a 10 cm Nunc EasYDish (Thermo Fisher Scientific) with 9 ml PYG and 10 μg ml$^{-1}$ of G418 (Nacalai Tesque). After 3 days of incubation, the medium was replaced with 10 ml of fresh PYG with 50 μg ml$^{-1}$ of G418 and incubated for 3 days. Subsequently, the cells were scraped with a cell lifter, and $5 \times 10^5$ cells were seeded into a 35 cm dish with 2 ml of PYG medium. APMV infection was performed as described above, and cells were collected with a cell lifter at 8 h.p.i. The cell suspension was centrifuged at 500 *g* for 5 min at room temperature. The cell pellets were lysed with 200 μl of 1× Passive Lysis Buffer (Promega) and stored at −80 °C until the luciferase-activity measurement. FLuc and RLuc activities were measured using the Dual-Luciferase Reporter Assay System (Promega) and the GloMax-Multi Detection System (Promega). FLuc activity was normalized by RLuc activity. Infection experiments were independently performed twice, and luciferase activity was measured in two technical replicates.

### Statistics and reproducibility

All statistical tests were done in R (v.4.3.1; R Core Team, 2023). Significance annotation in boxplots was done using the R package ggpubr (v.0.6.0; https://rpkgs.datanovia.com/ggpubr/). Exact *P* values for each subfigure are saved in Supplementary Tables 2 and 4. Representative images from over three independent biological replicates are shown (Fig. 5). Various conditions were tested once to optimize the tRNA-seq sequencing protocol (Extended Data Fig. 5).

No statistical methods were used to pre-determine sample sizes. The number of biological replicates for experiments were chosen based on previous studies and are similar to those generally used in the field. Data collection and analysis were not performed blind to the conditions of the experiments. Sequencing data were processed using standardized computational pipelines. There is no subjective scoring in our analysis.

### Reporting summary

Further information on research design is available in the Nature Portfolio Reporting Summary linked to this article.

### Data availability

The RNA-seq, Ribo-seq and mim-tRNA-seq results obtained in this study have been deposited in the NCBI Gene Expression Omnibus (GEO) database under the following accession numbers: RNA-seq, GSE276076; Ribo-seq, GSE276078; and mim-tRNA-seq, GSE276080. Source data are provided with this paper.

### Code availability

The key codes for the analyses of RNA-seq, Ribo-seq and mim-tRNA-seq are publicly available via Github (https://github.com/ruixuan-zhang/Zhang_Lotte_GiantVirusRiboSeq) and Zenodo (https://doi.org/10.5281/zenodo.16555392)[81].

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

## Acknowledgements

Computation time was provided by the Supercomputer System, Institute for Chemical Research, Kyoto University, and the HOKUSAI SailingShip supercomputer facility at RIKEN. This study was supported by the Japan Society for the Promotion of Science (JP23KJ1258 (to R.Z.), JP18H02279 (to H.O.), JP22H00384 (to H.O.) and JP24H02307 (to S.I.)), The Japan Science Society (Sasakawa Scientific Research Grant 2022-4103 (to R.Z.)), RIKEN (TRIP initiative 'TRIP-AGIS' to S.I.) and the Japan Science and Technology Agency (JPMJFS2123 (to R.Z.)). A.W. acknowledges funding from the European Union's Horizon 2020 research and innovation programme under the Marie Sklodowska-Curie grant agreement number 891572 and the European Union (European Research Council, CHIMERA, 101039843). Views and opinions expressed are, however, those of the author(s) only and do not necessarily reflect those of the European Union or the European Research Council Executive Agency. Neither the European Union nor the granting authority can be held responsible for them. APMV was provided by B. La Scola, Aix-Marseille Université, Marseille, France. R.Z. was a recipient of a fellowship from Japan Society for the Promotion of Science (DC2) and was supported by International Short-term Exchange Program for Young Researchers by the ICR-iJURC Kyoto University. We thank M. Biswas for editing a draft of this manuscript. We thank M. Imanishi for technical assistance on the reporter assay.

## Author contributions

Conceptualization, R.Z., L.M., H.H., A.W., S.I. and H.O.; methodology, M.M., A.W., Y.S. and S.I.; formal analysis, R.Z. and L.M.; investigation, R.Z., L.M., H.H., Y.S. and M.M.; writing—original draft, R.Z., L.M., S.I. and H.O.; writing—review and editing, R.Z., L.M., H.H., Y.S., M.M., A.W., S.I. and H.O.; visualization, R.Z. and L.M.; supervision, H.H., Y.S., A.W., S.I. and H.O.; project administration, S.I. and H.O.; funding acquisition, R.Z., A.W., S.I. and H.O.

## Competing interests

S.I. is a member of the *Scientific Reports* editorial board. The other authors declare no competing interests.

## Additional information

**Extended data** is available for this paper at https://doi.org/10.1038/s41564-025-02234-x.

**Correspondence and requests for materials** should be addressed to Shintaro Iwasaki or Hiroyuki Ogata.

**Extended Data Table 1 | Average tRNA supply (*W* score) for AU3 and GC3 codons in each sample and *p* value from Wilcoxon rank sum test (one-sided)**

| Sample | Codon type | | *p* value | |
|---|---|---|---|---|
| | AU3 | GC3 | AU3<GC3 | AU3>GC3 |
| R1 0 hpi | 2.1672 | 2.8443 | 0.0587 | 0.9429 |
| R2 0 hpi | 2.1429 | 2.8509 | 0.0450 | 0.9563 |
| R1 2 hpi | 2.1709 | 2.8497 | 0.0587 | 0.9429 |
| R2 2 hpi | 2.1463 | 2.8408 | 0.0457 | 0.9557 |
| R1 4 hpi | 2.1051 | 2.8212 | 0.0587 | 0.9429 |
| R2 4 hpi | 2.0956 | 2.8264 | 0.0464 | 0.9550 |
| R1 8 hpi | 2.1713 | 2.8126 | 0.0696 | 0.9323 |
| R2 8 hpi | 2.1598 | 2.8240 | 0.0580 | 0.9437 |

**Extended Data Table 2 | Average balance score for AU3 and GC3 codons in each sample and *p* value from Wilcoxon rank sum test (one-sided)**

| Sample | Codon type | | *p* value | |
|---|---|---|---|---|
| | **AU3** | **GC3** | **AU3<GC3** | **AU3>GC3** |
| R1 0 hpi | 0.3858 | 0.0991 | 1.0000 | 1.2017e-06 |
| R2 0 hpi | 0.3656 | 0.0979 | 1.0000 | 1.2017e-06 |
| R1 2 hpi | 0.3024 | 0.3227 | 2.5347e-01 | 7.5113e-01 |
| R2 2 hpi | 0.2568 | 0.3004 | 1.8112e-01 | 8.2265e-01 |
| R1 4 hpi | 0.2116 | 0.3451 | 9.1690e-03 | 9.9118e-01 |
| R2 4 hpi | 0.1773 | 0.3326 | 3.7242e-03 | 9.9643e-01 |
| R1 8 hpi | 0.1445 | 0.3236 | 5.2970e-04 | 9.9950e-01 |
| R2 8 hpi | 0.0796 | 0.3099 | 1.5285e-06 | 1.0000 |

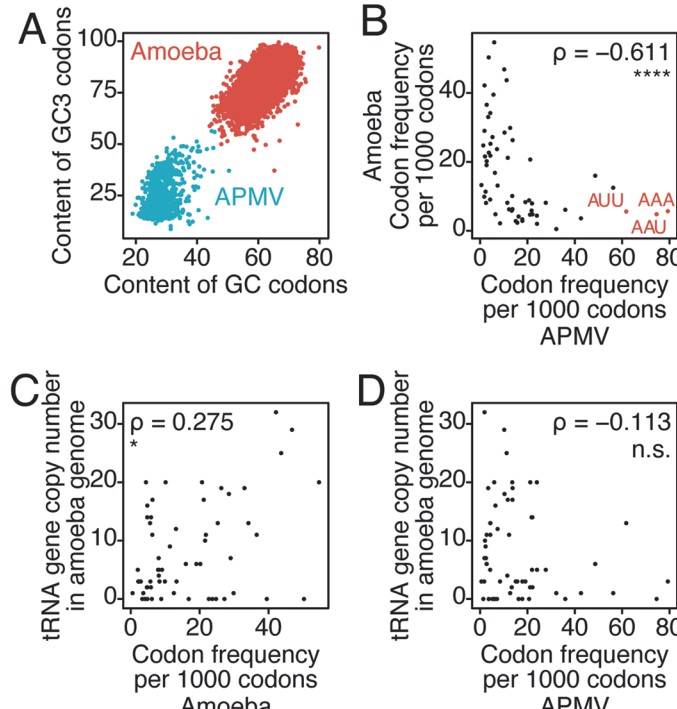

**Extended Data Fig. 1 | Codon usage difference between amoeba and APMV genomes.** (**a**) Correlation between G + C content of codons and G + C content of the 3rd position of codons in amoeba (red) and APMV (light blue) mRNAs. (**b**) Correlation between codon frequency per 1000 codons in amoeba and APMV mRNAs. (**c**) Correlation between codon frequency per 1000 codons of amoeba mRNAs and corresponding tRNA copy number in the amoeba genome. (**d**) Correlation between codon frequency per 1000 codons of APMV mRNAs and corresponding tRNA copy number in the amoeba genome. ρ, Spearman's correlation coefficient (two-tailed). ns, not significant; *, $p < 0.05$; **, $p < 0.01$; ***, $p < 0.001$; ****, $p < 0.0001$.

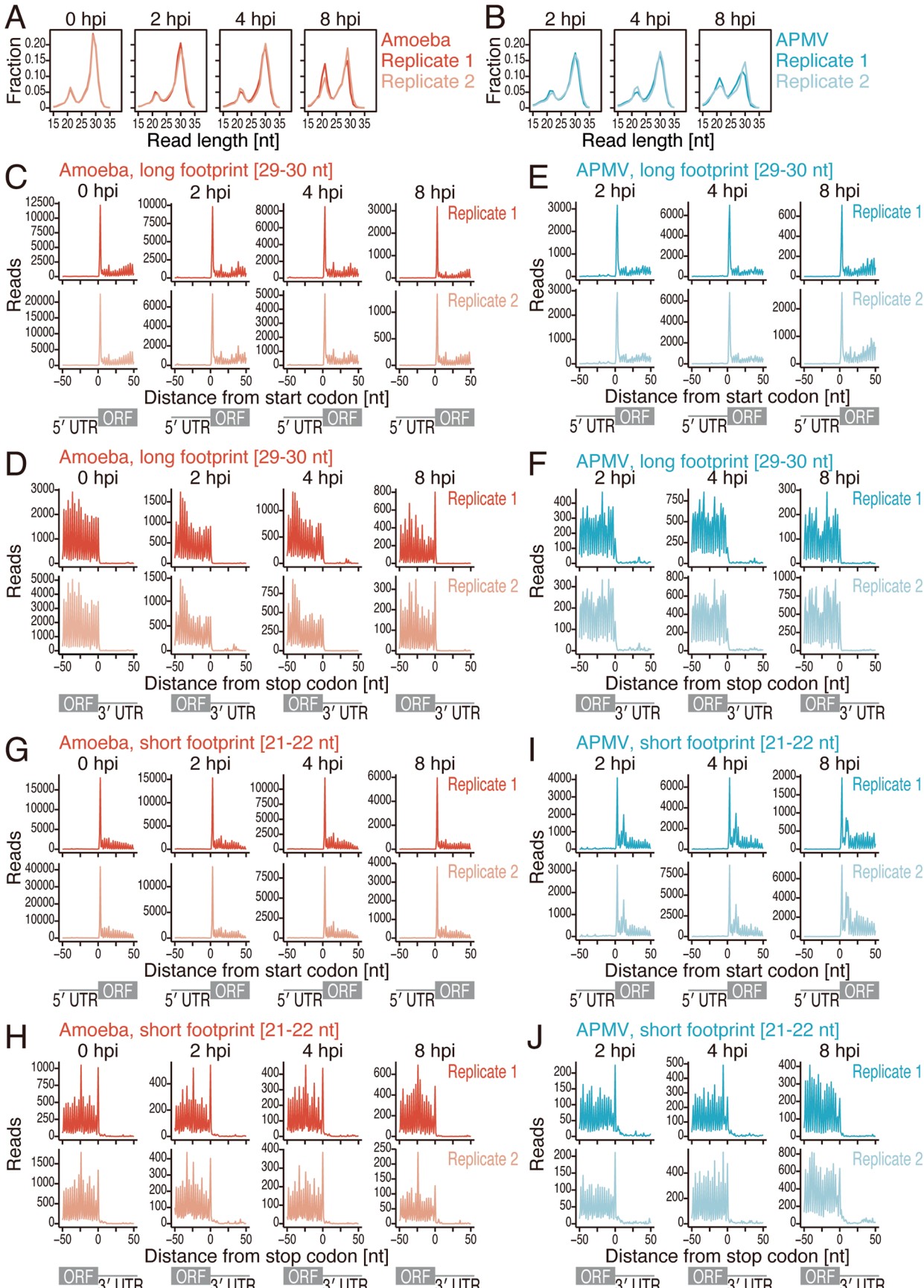

**Extended Data Fig. 2 | Ribosome footprints mapping around start and stop codon. (a, b)** Distribution of ribosome footprint length at different infection stages. Reads originating from amoeba and APMV mRNAs were analysed separately. hpi, hours post-infection. **(c–j)** Metagene plots of the relative footprint distribution around start codons (**c, e, g**, and **i**) and stop codons (**d, f, h**, and **j**) at different infection stages. Long (29-30 nt, **c–f**) and short (21-22 nt, **g–j**) footprints were analyzed separately. Reads originating from amoeba (**c, d, g**, and **h**) and APMV (**e, f, i**, and **j**) mRNAs were analysed separately.

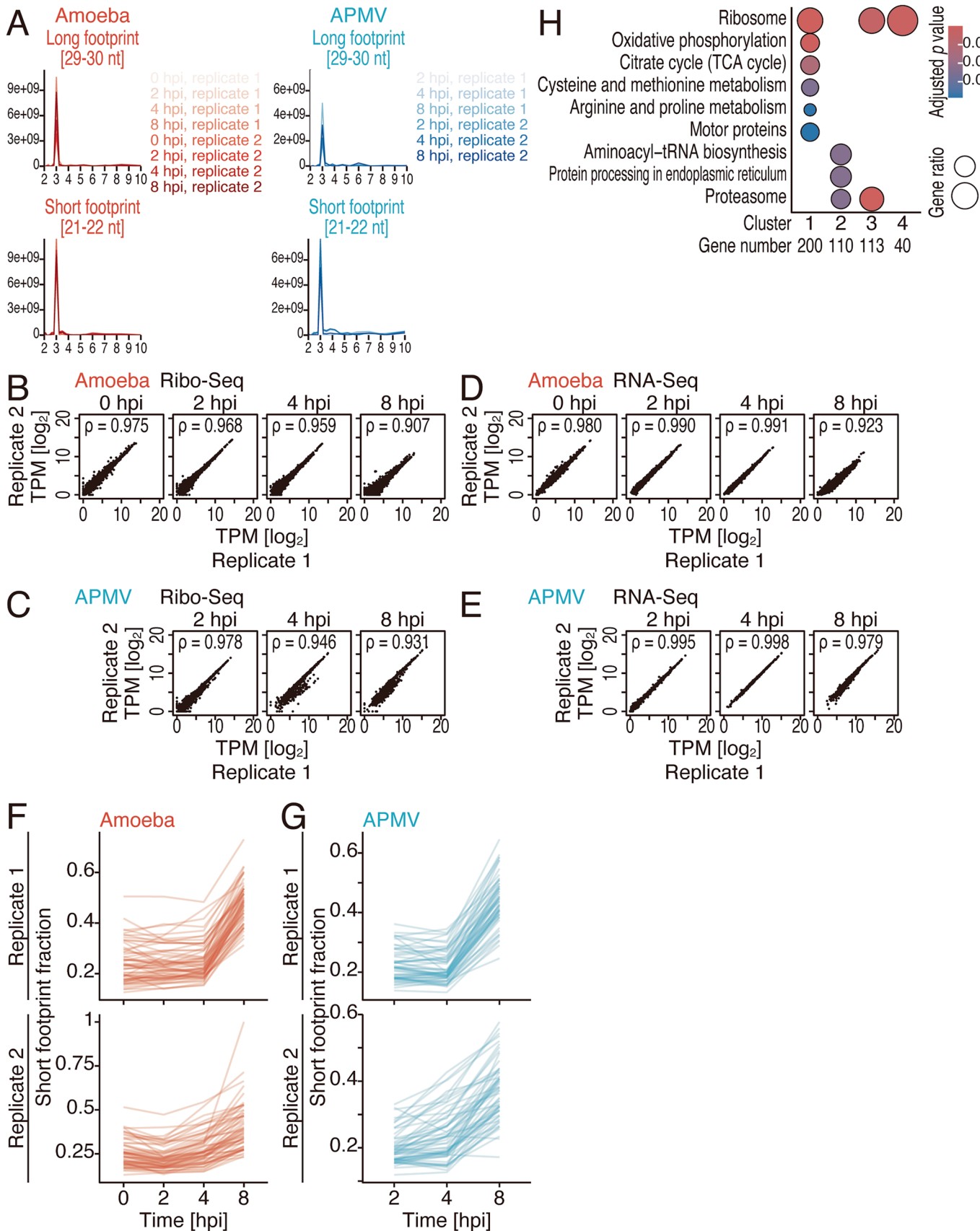

**Extended Data Fig. 3 | See next page for caption.**

**Extended Data Fig. 3 | Characterization of Ribo-Seq and RNA-Seq data.**
(**a**) Discrete Fourier transform of long (29-30 nt, top) and short (21-22 nt, bottom) footprints downstream of the start codon of the amoeba (left) and APMV (right) at different infection stages. (**b-e**) Scatter plots of the transcripts per kilobase million (TPM) values from two independent biological replicates for ribosomal footprints (**b, c**) and mRNAs (**d, e**) at different infection stages. (**f, g**) Short footprint accumulation during viral infection. Short footprint ratios on each codon type in amoeba (**f**) and APMV (**g**) mRNAs at different infection stages. (**h**) Enriched KEGG pathways in each host gene cluster. The colour scale for significance and the size scale for the gene ratio in each category are shown. The $p$ values were calculated by the hypergeometric distribution implemented in the enrichGO function in clusterProfiler. The Benjamini–Hochberg method was used to control the false discovery rate. The cutoffs for the $p$ value and $q$-value were both 0.05.

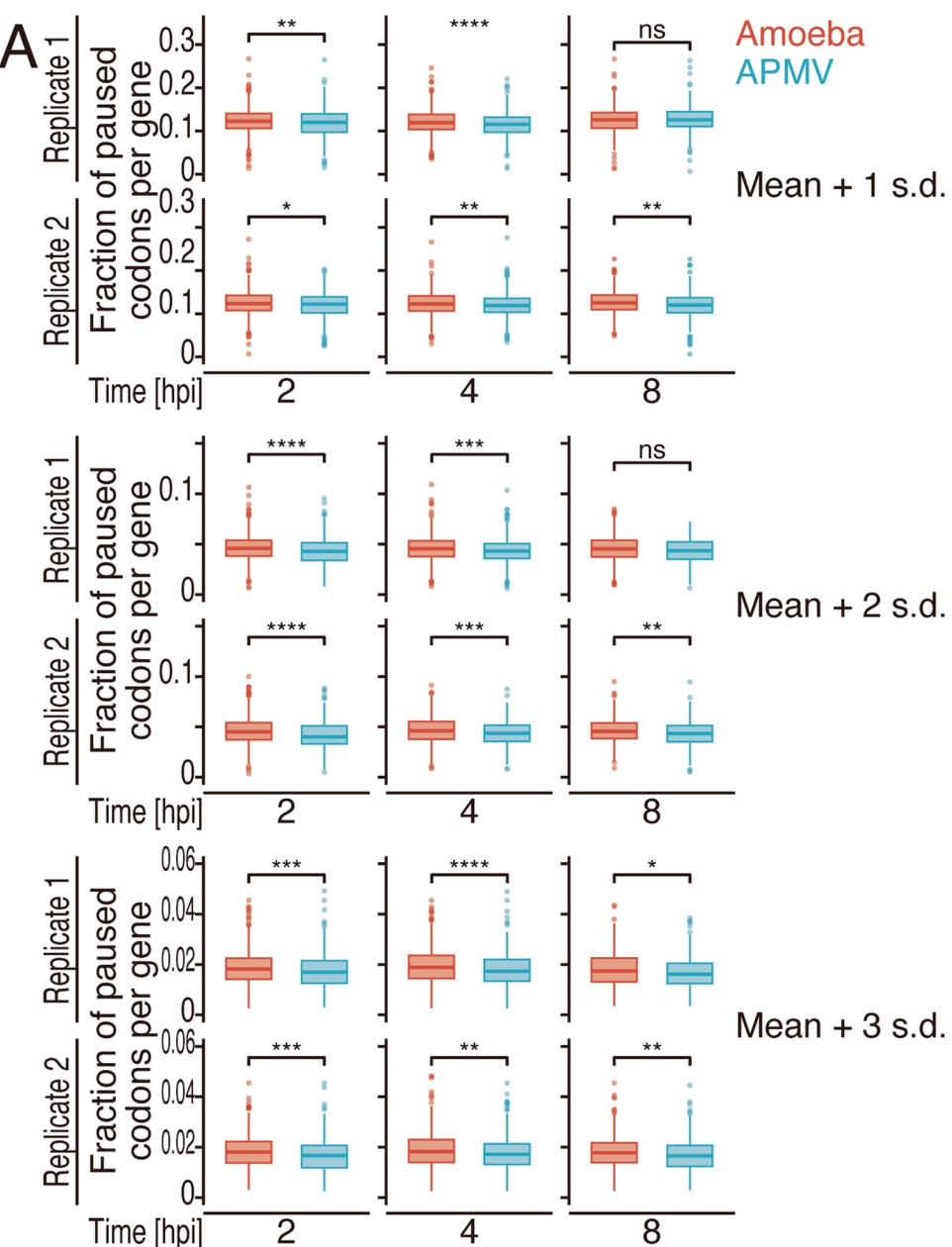

**Extended Data Fig. 4 | Detection of paused codons per gene. (a)** Box plots of the fraction of paused codons per gene in the amoeba and APMV genomes at different infection stages with different cutoffs for the paused codons. Note that the middle panels are the same plot as Fig. 2d. The median (centre line),

upper/lower quartiles (box limits), 1.5× interquartile range (whiskers), and outliers (points) are shown. The p values were calculated by the one-sided Wilcoxon rank sum test. ns, not significant; *, $p < 0.05$; **, $p < 0.01$; ***, $p < 0.001$; ****, $p < 0.0001$.

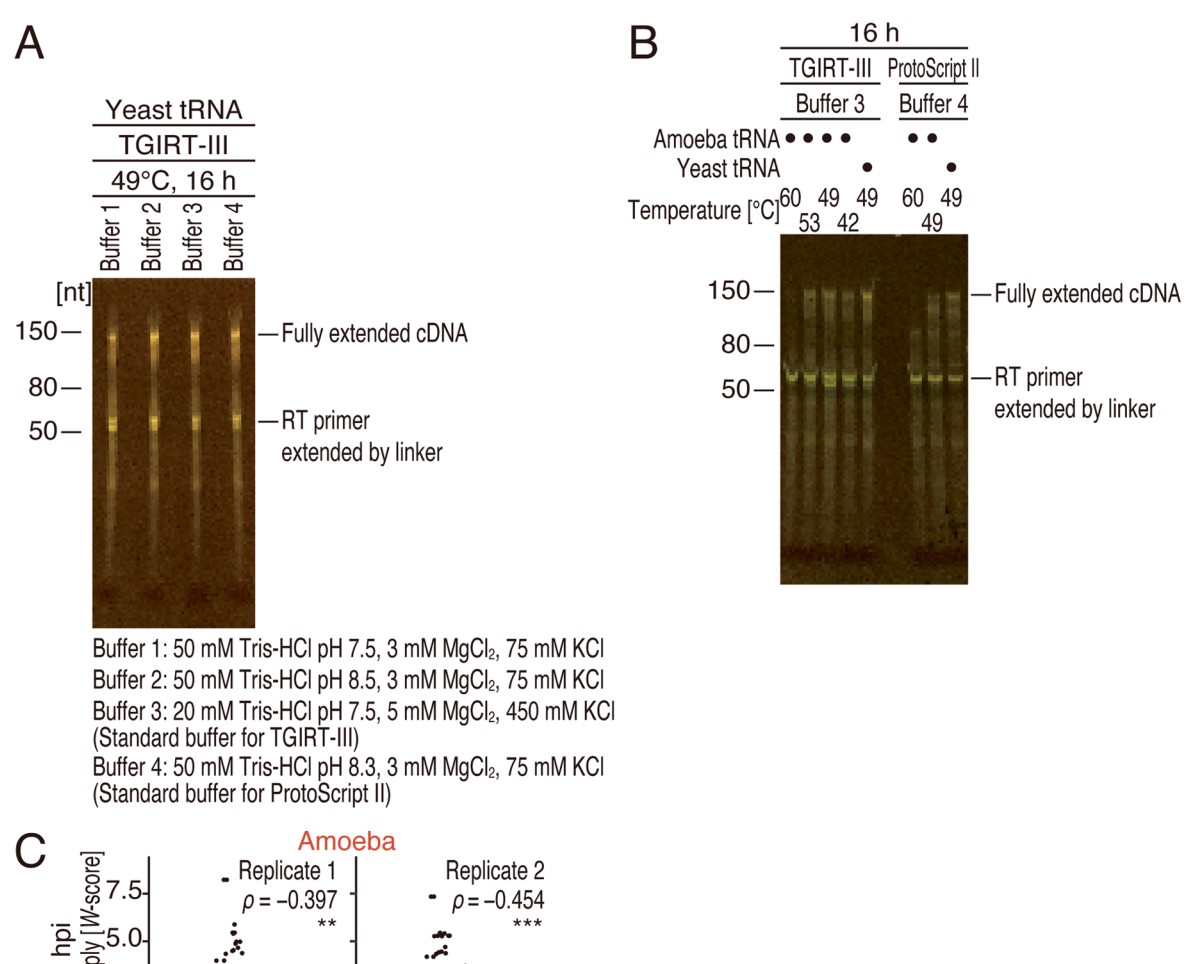

**A**

Yeast tRNA
TGIRT-III
49°C, 16 h

Buffer 1 | Buffer 2 | Buffer 3 | Buffer 4

[nt]
150 —
80 —
50 —

— Fully extended cDNA

— RT primer
extended by linker

Buffer 1: 50 mM Tris-HCl pH 7.5, 3 mM MgCl₂, 75 mM KCl
Buffer 2: 50 mM Tris-HCl pH 8.5, 3 mM MgCl₂, 75 mM KCl
Buffer 3: 20 mM Tris-HCl pH 7.5, 5 mM MgCl₂, 450 mM KCl
(Standard buffer for TGIRT-III)
Buffer 4: 50 mM Tris-HCl pH 8.3, 3 mM MgCl₂, 75 mM KCl
(Standard buffer for ProtoScript II)

**B**

16 h

TGIRT-III | ProtoScript II
Buffer 3 | Buffer 4

Amoeba tRNA ● ● ● ● ● ●
Yeast tRNA ● ●
Temperature [°C] 60 49 49 60 49
53 42 49

150 —
80 —
50 —

— Fully extended cDNA

— RT primer
extended by linker

**C**

Amoeba

Replicate 1
ρ = −0.397
**

Replicate 2
ρ = −0.454
***

0 hpi tRNA supply [W-score]
7.5
5.0
2.5
0

Ribosome occupancy
0 hpi

**Extended Data Fig. 5 | Optimization of library preparation for mim-tRNA-Seq.**
(**a, b**) Optimization of the reverse transcription condition was conducted using
titrating buffers, the reverse transcriptases TGIRT-III and ProtoScript II, and
temperatures. Linker-ligated tRNAs (from yeasts or amoebae) at the 3′ end
were used for the experiments. (**c**) Comparison of tRNA supply (W-score) and
ribosome occupancy for amoeba mRNAs in the growing condition. ρ, Spearman's
correlation coefficient (two-tailed). **, $p < 0.01$; ***, $p < 0.001$.

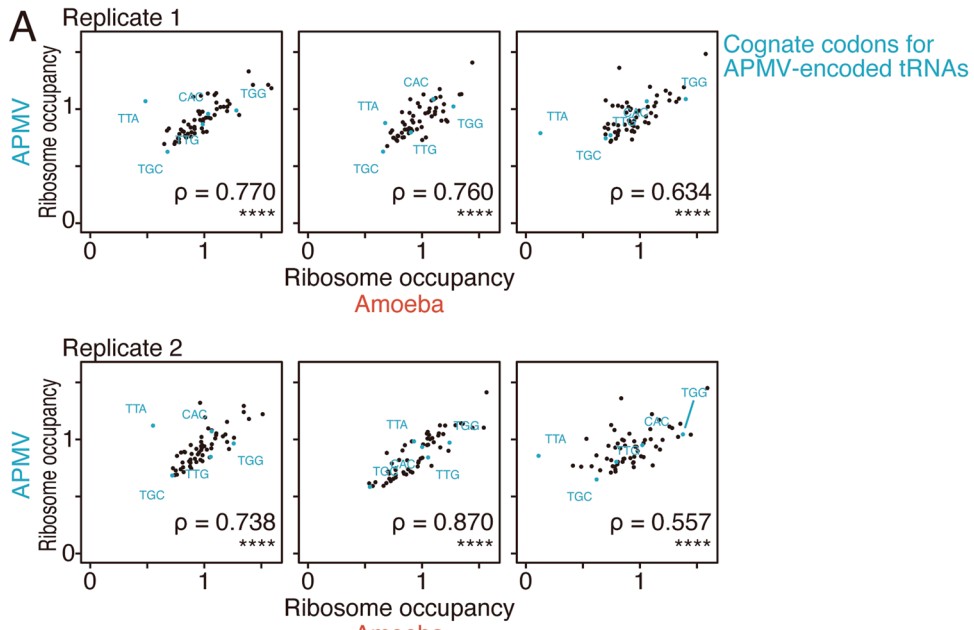

**Extended Data Fig. 6 | Correlation between amoeba and APMV ribosome occupancy. (a)** Scatter plots of the codon-specific ribosome occupancy on amoeba (horizontal axis) and APMV (vertical axis) mRNAs at different infection stages. Cognate codons of APMV-encoded tRNAs were coloured in pink. hpi, hours post-infection; ρ, Spearman's correlation coefficients (two-tailed); ****, $p < 0.0001$.

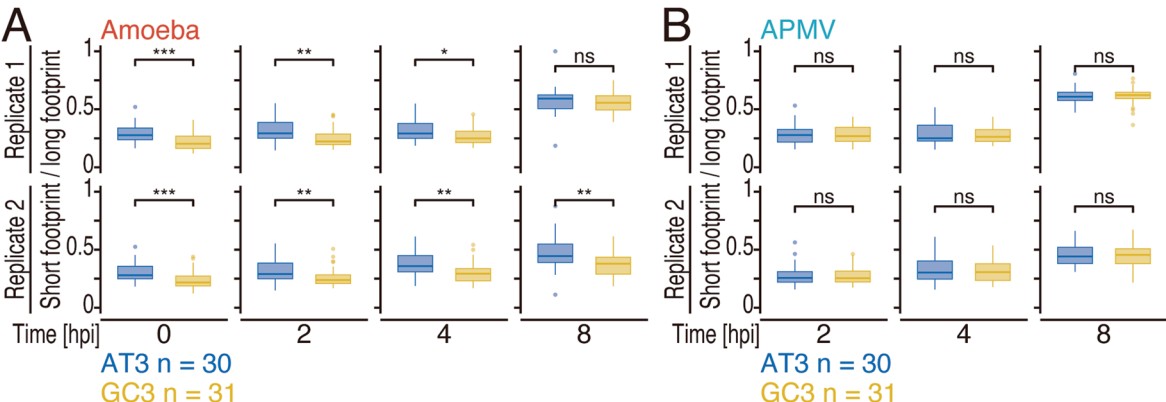

**Extended Data Fig. 7 | The short footprint/long footprint ratio at different infection stages.** (**a, b**) Box plots of the short footprint/long footprint ratio on AU3 and GC3 codons in amoeba (**a**) and APMV (**b**) mRNAs at different infection stages. AU3, codons ending with A or U; GC3, codons ending with G or C;

hpi, hours post-infection. The median (centre line), upper/lower quartiles (box limits), 1.5× interquartile range (whiskers), and outliers (points) are shown. The p values were calculated by the one-sided Wilcoxon rank sum test. ns, not significant; *, $p < 0.05$; **, $p < 0.01$; ***, $p < 0.001$; ****, $p < 0.0001$.

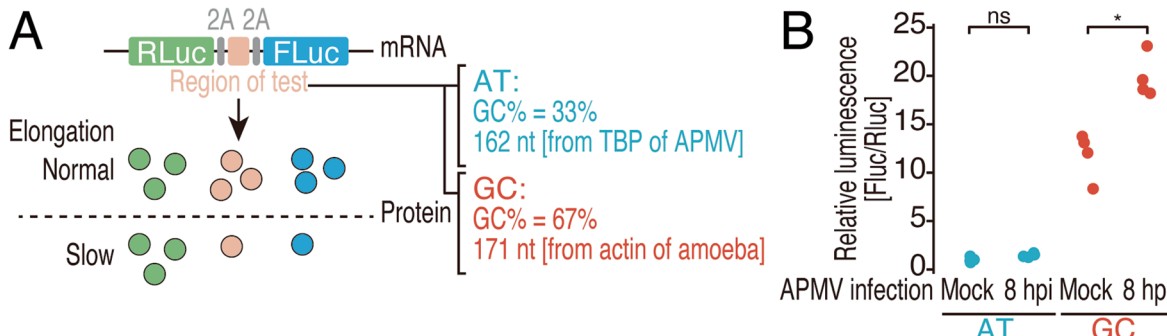

**Extended Data Fig. 8 | Reporter assay to monitor ribosome pausing during infection.** (**a**) Schematic of reporter mRNAs used in this study. Rluc, Renilla luciferase; Fluc, firefly luciferase. (**b**) Relative Fluc/Rluc luminescence (the average values of the AT repoter with mock infection was set as 1). Individual data points (n = 4) are shown. The p values were calculated by the one-sided Wilcoxon rank sum test. ns, not significant; *, $p < 0.05$.

# Reporting Summary

## Statistics

For all statistical analyses, confirm that the following items are present in the figure legend, table legend, main text, or Methods section.

| n/a | Confirmed | |
|---|---|---|
| ☐ | ☒ | The exact sample size (*n*) for each experimental group/condition, given as a discrete number and unit of measurement |
| ☐ | ☒ | A statement on whether measurements were taken from distinct samples or whether the same sample was measured repeatedly |
| ☐ | ☒ | The statistical test(s) used AND whether they are one- or two-sided *Only common tests should be described solely by name; describe more complex techniques in the Methods section.* |
| ☒ | ☐ | A description of all covariates tested |
| ☐ | ☒ | A description of any assumptions or corrections, such as tests of normality and adjustment for multiple comparisons |
| ☐ | ☒ | A full description of the statistical parameters including central tendency (e.g. means) or other basic estimates (e.g. regression coefficient) AND variation (e.g. standard deviation) or associated estimates of uncertainty (e.g. confidence intervals) |
| ☐ | ☒ | For null hypothesis testing, the test statistic (e.g. $F$, $t$, $r$) with confidence intervals, effect sizes, degrees of freedom and $P$ value noted *Give P values as exact values whenever suitable.* |
| ☒ | ☐ | For Bayesian analysis, information on the choice of priors and Markov chain Monte Carlo settings |
| ☒ | ☐ | For hierarchical and complex designs, identification of the appropriate level for tests and full reporting of outcomes |
| ☒ | ☐ | Estimates of effect sizes (e.g. Cohen's *d*, Pearson's *r*), indicating how they were calculated |

*Our web collection on statistics for biologists contains articles on many of the points above.*

## Software and code

Policy information about availability of computer code

| | |
|---|---|
| Data collection | Ribosome profiling, RNA-Seq, and tRNA-Seq data were collected using HiSeq X (Illumina) |
| Data analysis | fastp v.0.21.0, STAR v.2.7.0a, SAMtools v.1.10, fastqc v.0.12.1, fastx-trimmer v.0.0.14, Bowtie2, RNAmmer v.1.2, tRNAscan-SE v.2.0.12, factoextra v.1.0.7, eggNOG-mapper webserver v.2.1.1.2, clusterProfiler v.4.8.3, ggpubr v.0.6.0, R v.4.3.1. <br><br> For quantification of ribosomal footprints and RNA-Seq reads, a custom script was used (https://github.com/ingolia-lab/RiboSeq) <br><br> Scripts for analysis and figures of high-resolution have been uploaded to zenodo under doi: 10.5281/zenodo.16555392 |

For manuscripts utilizing custom algorithms or software that are central to the research but not yet described in published literature, software must be made available to editors and reviewers. We strongly encourage code deposition in a community repository (e.g. GitHub). See the Nature Portfolio guidelines for submitting code & software for further information.

## Data

Policy information about availability of data

All manuscripts must include a data availability statement. This statement should provide the following information, where applicable:
  - Accession codes, unique identifiers, or web links for publicly available datasets
  - A description of any restrictions on data availability
  - For clinical datasets or third party data, please ensure that the statement adheres to our policy

> The RNA-Seq, Ribo-Seq, and mim-tRNA-Seq results obtained in this study have been deposited in the National Centre for Biotechnology Information (NCBI) database under following accession numbers.
>
> RNA-Seq: GSE276076 (https://www.ncbi.nlm.nih.gov/geo/query/acc.cgi?acc=GSE276076)
> Ribo-Seq: GSE276078 (https://www.ncbi.nlm.nih.gov/geo/query/acc.cgi?acc=GSE276078)
> tRNA-Seq: GSE276080 (https://www.ncbi.nlm.nih.gov/geo/query/acc.cgi?acc=GSE276080)

## Research involving human participants, their data, or biological material

Policy information about studies with human participants or human data. See also policy information about sex, gender (identity/presentation), and sexual orientation and race, ethnicity and racism.

| | |
|---|---|
| Reporting on sex and gender | Not applicable |
| Reporting on race, ethnicity, or other socially relevant groupings | Not applicable |
| Population characteristics | Not applicable |
| Recruitment | Not applicable |
| Ethics oversight | Note applicable |

Note that full information on the approval of the study protocol must also be provided in the manuscript.

# Field-specific reporting

Please select the one below that is the best fit for your research. If you are not sure, read the appropriate sections before making your selection.

☒ Life sciences  ☐ Behavioural & social sciences  ☐ Ecological, evolutionary & environmental sciences

For a reference copy of the document with all sections, see nature.com/documents/nr-reporting-summary-flat.pdf

# Life sciences study design

All studies must disclose on these points even when the disclosure is negative.

| | |
|---|---|
| Sample size | The study focuses on investigating the translation dynamics during the infection course of acanthamoeba castellanii mimivirus. We conducted a time-course virus infection experiments and collected samples at 0 (for mock infection), 2, 4, and 8 hours post infection. For each time point, two independent biological replicates were prepared. When conducting statistical test at gene level, all genes with enough reads were used. For codon-wise comparison, all codons except stop codon were used. Detailed information about the sample numbers is provided in the manuscript. These samples were used for ribosome profiling, RNA-Seq and tRNA-Seq, in line with the standard protocols in the field. No statistical methods were employed to predetermine the sample size. The sample size is similar to those generally employed in the field. |
| Data exclusions | No data were excluded from the analysis. |
| Replication | The reproducibility of our experiments was confirmed with two replicates tested for deep sequencing experiments. All experiments were repeatable. Results can be reproduced with the parameters and program provided in the methods section. |
| Randomization | Randomization is not relevant to this study. |
| Blinding | Blinding is not applicable to this study. |

# Reporting for specific materials, systems and methods

We require information from authors about some types of materials, experimental systems and methods used in many studies. Here, indicate whether each material, system or method listed is relevant to your study. If you are not sure if a list item applies to your research, read the appropriate section before selecting a response.

## Materials & experimental systems

| n/a | Involved in the study |
|-----|----------------------|
| ☒ | ☐ Antibodies |
| ☐ | ☒ Eukaryotic cell lines |
| ☒ | ☐ Palaeontology and archaeology |
| ☒ | ☐ Animals and other organisms |
| ☒ | ☐ Clinical data |
| ☒ | ☐ Dual use research of concern |
| ☒ | ☐ Plants |

## Methods

| n/a | Involved in the study |
|-----|----------------------|
| ☒ | ☐ ChIP-seq |
| ☒ | ☐ Flow cytometry |
| ☒ | ☐ MRI-based neuroimaging |

## Eukaryotic cell lines

Policy information about cell lines and Sex and Gender in Research

| | |
|---|---|
| Cell line source(s) | Acanthamoeba castellanii, strain Neff [American Type Culture Collection (ATCC), 30010] |
| Authentication | n.a |
| Mycoplasma contamination | n.a |
| Commonly misidentified lines (See ICLAC register) | n.a |

## Plants

| | |
|---|---|
| Seed stocks | n.a |
| Novel plant genotypes | n.a |
| Authentication | n.a |

