## [Peer Review File · Nature Microbiology]

A giant virus forms a specialized subcellular environment within its amoeba host for efficient translation

Corresponding Author: Professor Hiroyuki Ogata

Version 0:

Decision Letter:

7th January 2025

Dear Professor Ogata,

Thank you for your patience while your manuscript "Giant virus creates subcellular environment to overcome codon-tRNA mismatch" was under peer-review at Nature Microbiology. It has now been seen by 3 referees, whose expertise and comments you will find at the end of this email. Although they find your work of some potential interest, they have raised a number of concerns that will need to be addressed before we can consider publication of the work in Nature Microbiology.

As you will see the referees share a number of technical concerns, in particular related to the sequencing (which should be cleared up with additional controls and checks) and the statistical support for some of the analyses. Should further experimental data allow you to address these criticisms, we would be happy to look at a revised manuscript.

Please include a data availability statement as a separate section after Methods but before references, under the heading "Data Availability". This section should inform readers about the availability of the data used to support the conclusions of your study. This information includes accession codes to public repositories (data banks for protein, DNA or RNA sequences, microarray, proteomics data etc...), references to source data published alongside the paper, unique identifiers such as URLs to data repository entries, or data set DOIs, and any other statement about data availability. At a minimum, you should include the following statement: "The data that support the findings of this study are available from the corresponding author upon request", mentioning any restrictions on availability. If DOIs are provided, we also strongly encourage including these in the Reference list (authors, title, publisher (repository name), identifier, year). For more guidance on how to write this section please see: <http://www.nature.com/authors/policies/data/data-availability-statements-data-citations.pdf>

* If you have not done so already we suggest that you begin to revise your manuscript so that it conforms to our Article format instructions at <http://www.nature.com/nmicrobiol/info/final-submission>. Refer also to any guidelines provided in this letter.

When submitting the revised version of your manuscript, please pay close attention to our [href="https://www.nature.com/nature-](https://www.nature.com/nature-)

portfolio/editorial-policies/image-integrity">Digital Image Integrity Guidelines. and to the following points below:

EXTENDED DATA FIGURES

Link Redacted

Note: This url links to your confidential homepage and associated information about manuscripts you may have submitted or be reviewing for us. If you wish to forward this e-mail to co-authors, please delete this link to your homepage first.

Nature Microbiology is committed to improving transparency in authorship. As part of our efforts in this direction, we are now requesting that all authors identified as 'corresponding author' on published papers create and link their Open Researcher and Contributor Identifier (ORCID) with their account on the Manuscript Tracking System (MTS), prior to acceptance. This applies to primary research papers only. ORCID helps the scientific community achieve unambiguous attribution of all scholarly contributions. You can create and link your ORCID from the home page of the MTS by clicking on 'Modify my Springer Nature account'. For more information please visit www.springernature.com/orcid.

If you wish to submit a suitably revised manuscript we would hope to receive it within 6 months. If you cannot send it within this time, please let us know. We will be happy to consider your revision, even if a similar study has been accepted for publication at Nature Microbiology or published elsewhere (up to a maximum of 6 months).

Yours sincerely,

Kyle

Dr. Kyle Frischkorn
(he/him/his)
Senior Editor, Nature Microbiology
Nature Portfolio

Reviewer Expertise:

- Referee #1: giant viruses
- Referee #2: RNAseq techniques, bioinformatics, virology
- Referee #3: giant viruses, sequencing, molecular biology

Reviewer Comments:

Reviewer #1 (Remarks to the Author):

Zhang et al conducted a study of transfer RNA pools and codon usage for the pair composed of Mimivirus and its host cell, the amoeba *Acanthamoeba castellanii*.

The discordance between the codon usage of the virus and the amoeba, and the existence of elements of the translation apparatus, including transfer RNAs encoded by the Mimivirus genome, are intriguing elements. This definitely deserves to be investigated in depth.

The authors of this manuscript used many and varied techniques and protocols to analyze different aspects of codon usage and transfer RNA pools within the pair Mimivirus and *Acanthamoeba* sp.

This study is well conducted, ordered in coherent steps. It provides interesting data to decipher some of the aspects of replication related to the mentioned objectives. However, some would in my opinion deserve to be further explained and discussed.

Line 231-232: The authors mention similar data obtained in influenza viruses and SARS-CoV-2: "Similar results have been reported during infections with influenza A virus and coronavirus SARS-CoV-2". These two viruses are very different from Mimivirus in terms of size and complexity. What can the authors take away from such a comparison?

Line 239: The authors should further explain this sentence and interesting point: "..., the preference for the AT-rich genome in APMV may be an adaptive strategy".

The last sentence of the discussion, lines 245-246: "This translation-associated bottleneck may drive the continuous acquisition of translation-related genes by the AT-rich genomes of APMV-related viruses", would also deserve to be further explained, because it describes an intriguing putative mechanism.

Authors should cite and comment on data from other articles cited below that have described codon usage and transfer RNA interplay in other viruses, including the points mentioned below:

- Serrano-Solís V, Toscano Soares PE, de Farías ST. Genomic Signatures Among *Acanthamoeba polyphaga* Entoorganisms Unveil Evidence of Coevolution. *J Mol Evol*. 2019 Jan;87(1):7-15. doi: 10.1007/s00239-018-9877-1. Epub 2018 Nov 20. PMID: 30456441: "The genomic signatures in these entoorganisms of diverse evolutionary origin are coevolutionarily conserved within an intracellular environment that provides sanctuary for species of ecological and biomedical relevance."

- Colson P, Fournous G, Diene SM, Raoult D. Codon usage, amino acid usage, transfer RNA and amino-acyl-tRNA synthetases in Mimiviruses. *Intervirology*. 2013;56(6):364-75. doi:10.1159/000354557. Epub 2013 Oct 17. PMID: 24157883: "..., we found that the genes most highly expressed at the beginning of the Mimivirus replicative cycle have a nucleotide content more adapted to the codon usage in *A.castellanii*".

- Michely S, Toulza E, Subirana L, John U, Cognat V, Maréchal-Drouard L, Grimsley N, Moreau H, Piganeau G. Evolution of codon usage in the smallest photosynthetic eukaryotes and their giant viruses. *Genome Biol Evol*. 2013;5(5):848-59. doi:10.1093/gbe/evt053. PMID: 23563969; PMCID: PMC3673656: "We show that 1) Codon usage bias in the host and in the viral genes increases with expression levels and 2) optimal codons use those tRNAs encoded by the most abundant host tRNA genes, supporting the notion of translational optimization by natural selection. We find evidence that viral tRNA genes complement the host tRNA pool for those viral amino acids whose host tRNAs are in short supply."

- Sau K, Gupta SK, Sau S, Mandal SC, Ghosh TC. Factors influencing synonymous codon and amino acid usage biases in Mimivirus. *Biosystems*. 2006 Aug;85(2):107-13. doi: 10.1016/j.biosystems.2005.12.004. Epub 2006 Jan 24. PMID: 16442213: "It was found that codon usage bias in Mimivirus genes is dictated both by mutational pressure and translational selection. Evidences show that four factors such as mean molecular weight (MMW), hydrophathy, aromaticity and cysteine content are mostly responsible for the variation of amino acid usage in Mimivirus proteins. Based on our observation, we suggest that genes involved in translation, DNA repair, protein folding, etc., have been laterally transferred to Mimivirus a long ago from living organism and with time these genes acquire the codon usage pattern of other Mimivirus genes under selection pressure."

- Finally, an article on bacteriophages: Bailly-Bechet M, Vergassola M, Rocha E. Causes for the intriguing presence of tRNAs in phages. *Genome Res*. 2007 Oct;17(10):1486-95. doi: 10.1101/gr.6649807. Epub 2007 Sep 4. PMID: 17785533; PMCID: PMC1987346: "Thus, even though phages use most of the cell's translation machinery, they can complement it with their own genetic information to attain higher fitness. These results suggest that similar selection pressures may act upon other cellular essential genes that are being found in the recently uncovered large viruses."

Line 90 to 95: the proportion of viral mRNAs varies significantly between replicates (62 and 85%, 40 and 81%). How do the authors explain these differences?

Lines 205-206: "Thus, our data suggest that Mimivirus created a unique environment tailored for viral mRNA translation at the periphery of the virus factory.": Is this environment physically delimited? How could it be?

The authors should comment on whether they think that their observations could also be found for other viruses for which a virus factory is formed during the replication cycle, referring for example to the following articles on virus factories:

- Netherton CL, Wileman T. Virus factories, double membrane vesicles and viroplasm generated in animal cells. *Curr Opin Virol*. 2011 Nov;1(5):381-7. doi: 10.1016/j.coviro.2011.09.008. Epub 2011 Oct 12. PMID: 22440839; PMCID: PMC7102809.

- Novoa RR, Calderita G, Arranz R, Fontana J, Granzow H, Risco C. Virus factories: associations of cell organelles for viral replication and morphogenesis. *Biol Cell*. 2005 Feb;97(2):147-72. doi: 10.1042/BC20040058. PMID: 15656780; PMCID: PMC7161905.

The authors should comment more on what the implications of their data and their interpretation might be?
Has such a compartment been already observed in other contexts with other microorganisms involved?
Does it exist for other microorganisms and viruses? Is it new?

Reviewer #1 (Remarks on code availability):

Possibly the link to the code (10.5281/zenodo.13748128) was not made usable yet as it does not work, or I did not manage to find the path.

Reviewer #2 (Remarks to the Author):

The manuscript involves an RNA-seq/Ribo-seq/tRNA-seq study of mimivirus-infected amoeba, backed up with microscopy including in situ probing of RNA and protein synthesis. Overall the experimental and bioinformatic work appear to be of a high standard, and the manuscript is clearly written and easy to follow. However there are some potential issues that need addressing

as outlined below.

Major comments:

o I don't think the title "Giant virus creates subcellular environment to overcome codon-tRNA mismatch" is supported by the data. Yes, APMV creates a subcellular environment - the viral factory (VF), as already known. But there is no direct evidence that this is "to overcome codon-tRNA mismatch". It was shown that ribosomes are recruited to the periphery of the VF, and translate viral mRNA there, but no direct evidence was presented that the local tRNA populations are altered compared to elsewhere in the cell. It's a nice story, but the evidence presented from the Ribo-seq analysis is indirect. Similarly, the statement on lines 236-238 ("APMV alleviates this translation-associated bottleneck by ..." should be written as a conjecture ("APMV may alleviate ...").

o One possible alternative explanation might be: "Virus mRNAs are AU rich which means they will have less RNA structure than the host GC-rich mRNAs. Less structured mRNAs might be translated more easily by amoeba ribosomes. So maybe lack of structure compensates for lack of codon optimality, rather than the viral factories altering local tRNA abundances?"

o The conclusions are critically dependent on several Ribo-seq analyses, including ribosome occupancy on codons, fraction of paused codons per gene, and the short/total footprint ratio. Thus it is essential that these measures are not substantially affected by artifacts. For reasons outlined below, further verification of this is needed before the results can be properly assessed. Of particular concern is the possibility of ribosome run-on occurring during sample preparation. As analysed by Hussmann et al 2015 (PMID 26656907), ribosome run-on can be a real issue, especially when using cycloheximide treatment without flash freezing (as the authors appear to have done) and this would have the effect of making meaningless any codon-specific statistics such as the above. Run-on effects can be sensitive to precise details of sample preparation, besides the cell type/species, so the authors' study may be OK. However it is critically important to bioinformatically and/or experimentally verify that run-on has not occurred in the authors' datasets. The analyses outlined in Hussmann et al 2015 could be a good place to start.

o In the analysis of ribosome pausing (lines 130-135), peaks in ribosome protected fragment (RPF) density were defined as pause sites. However, RPF peaks can also result from biases (ligation bias, nuclease bias, PCR bias). More work is needed to determine whether the "mean + 2 standard deviations" threshold is sufficient to select for pause peaks over bias peaks. One possible way forward would be to investigate correlation between RPF peaks and A/P-site codons relative to correlation between RPF peaks and the 5'/3' terminal nucleotides that would have the strongest effect on ligation bias. There is relevant discussion in O'Connor et al 2016 (PMID 27698342). Particularly in the scenario presented in this manuscript, one cannot assume that "bias will average out": because the virus mRNAs are AU-rich and the host mRNAs are GC-rich, they will be subject to very different biases.

o I'm actually surprised the authors saw any of the short (21-22 nt) RPFs without using flash freezing or the translation inhibitor drugs that others have used to reveal this conformation of the ribosome. There are difference between e.g. yeast and mammalian cells so it may be that this is a reflection of the different system (amoeba). Measuring the short/total footprint ratio is a nice idea for aiming to quantify tRNA accessibility. But I have a few comments

- Sorry if I missed it, but is it specified that this analysis is carried out for the A-site (not P-site) codon?

- In the "Short footprint ratio" section in Methods, please specify which read lengths are used for "short" and for "total" number of footprints. Would the measure be cleaner if it was just 21-22 nt versus 29-30 nt reads?

- In Fig 1C of Cook et al 2022 (PMID 35226596), short and long footprints were detected at different stages of PRRSV virus infection, but the short/total ratio doesn't seem to be well-correlated with hpi; instead it seems more likely to be rather sensitive to small differences in sample preparation. The authors' own sample preparation may be more robust than the Cook et al study in this respect, but they may like to consider whether possible variation in sample preparation could impact on their data.

- In nearly all Ribo-seq studies, there is some amount of reads that map to mRNAs but do not derive from bona fide RPFs, e.g. they might derive from other RNP complexes co-sedimenting with ribosomes. If nuclease trimming works exceptionally well then an upper limit can be put on non-RPF contamination by looking at read-length plots (Ext Fig 2A,B) and triplet phasing (proportion of reads mapping to 1st, 2nd and 3rd positions of codons). Even if nuclease trimming doesn't work so well, an upper limit can be put on non-RPF contamination by looking at 3'UTR:CDS density ratio statistics. For example, if this was 1% for mock but 15% for 8 hpi, then one would infer that non-RPF contamination was increasing with hpi and might be 15% of the total density at 8 hpi. In the authors' Ext Fig 2A,B, the plots will represent true short RPFs, true long RPFs, and a background of non-RPF contamination. If this non-RPF contamination increases with hpi (as is often the case in Ribo-seq studies of other viruses) then it will contribute to the changing short/total ratio, potentially explaining some or all of the observed increase. Again, as in Ribo-seq studies of other viruses, the increase in non-RPF contamination at later time points is generally different for host compared to virus, and - depending on the virus species - can affect host mRNAs more or can affect virus mRNAs more. It is important that the authors put bounds on how these possibilities might affect the short/total ratio used in the manuscript. As a start, one could (i) use the 3'UTR density (provided virus and host 3'UTRs are long enough) to put an upper limit on contamination density at each time point; and (ii) compare virus with host triplet phasing at each timepoint, separately for the 21-22 nt and 29-30 nt reads. I'd also like to see "distance from start codon" (like Ext Fig 2C,D) and "distance from stop codon" plots separately for the 21-22 nt and 29-30 nt reads (e.g. to verify the 21-22 nt reads are RPFs and not miRNAs/siRNAs :-)).

These last two points could help address some of the above points, but may be impracticable and I put them here just as suggestions:

o Ideally, the manuscript would include a positive control for what Ribo-seq codon occupancy/pausing looks like when a tRNA species is in short supply (cf. the Wu et al 2019 paper, PMID 30686592, where they used 3-amino-1,2,4-triazole (3-AT) treatment

to inhibit histidine biosynthesis and deplete histidyl-tRNA in yeast as a positive control for pausing at A-site His codons). I realise that this may not be a very practicable question to address for a non-model organism, but it would strengthen claims such as (lines 159-160) "smooth elongation of GC3 codons" implying "no shortage of tRNA supply".

o The manuscript contends that the viral factories (or at least something about virus infection) is responsible for making viral mRNAs efficiently translated despite having codon usage that is very different from host. Is it possible to transfect/overexpress several viral mRNAs in the absence of viral infection, ribosome profile, and assess whether - in contrast to the same mRNAs in virus infection - they exhibit poor translation, ribosome pausing, etc? I realise this may not be practicable (transfection of a non-model organism, difficulty getting high enough expression to get enough Ribo-seq coverage, cost).

Reviewer #3 (Remarks to the Author):

The authors of the manuscript entitled: "Giant virus creates subcellular environment to overcome codon-tRNA mismatch" present a compelling case of host cell manipulation by a virus, focusing on the Mimiviridae infecting Amoebas. In general, the manuscript is very well written and structured, the materials and methods go in great detail and the figures are of appropriate pixel density and colours to make them readable and understandable. The authors employed a plethora of molecular and cell biology techniques on their investigation, presenting a well rounded hypothesis.

As I already mentioned, the manuscript is quite well written and structured. However, there are some terms and concepts that are introduced at a rather late stage. A prime example I noticed was the definition of the "footprints" and their significance, that are already mentioned in the introduction and results, but are only defined later. I strongly suggest that the authors introduce the term already in the introduction.

In the materials and methods it would be beneficial to write how much raw sequencing data was produced per sample or per investigation (e.g. line 288: "...pair-end read option, and a total yield of XX gigabases). All other methods are described in great extent.

My major comments are on the conclusions and on the Boxplots. If we start with the boxplots (especially Figures 1 and 2) it seems that there is something going on with the statistics. In particular, if we take fig 1D, 1E and 1H as examples, there is suggested to be a very strong statistical significance in the differences between Amoeba and APMV, even though it does not look like from the distribution of observations. Since the code is not available yet, is this an artifact of the analysis that puts too much weight on the actual number of observations (Amoeba: 10575, APMB 955) which would overshadow any other effect? The same applies for Figure 2. Fig 3A and 3B have similar numbers of observations between host and virus and there there is no statistically significant difference, while for Fig 3D there is an effect, that again may be affected just by the number of observations. Code availability to the reviewers would be able to answer this.

My last remark is regarding the suggestion of a microcompartment in the host's cell that allows the virus to snatch all available AU3-related tRNAs. I was of course happy to read that the authors suggest an evolution of the virus to make use of the high GC preference of the host, but from Extended Data Fig.1A there seem to be virtually no Amoeba codons that have GC content below 50%. When I read this figure I see basically zero competition between host and virus, so the fact that ribosomes are close to the VF and that viral mRNA translation proceeds happily may just be due to the fact the host does not need these tRNAs at all so they naturally are found closer to the VF upon infection. The authors also suggest that synthesis of viral proteins is limited by tRNA availability because of the availability of viral mRNAs and the ration of AT-rich to GC-rich tRNAs in the host. For that to be viable there would need to be data showing either some kind of transcription regulation in viruses, translation kinetics in the presence of abundant tRNAs or an effect of different concentrations of the right tRNAs and the number of viral particles.

Reviewer #3 (Remarks on code availability):

The Zenodo archive that hosts the code is not yet publicly available. The same applies for the NCBI repositories that host the raw sequencing data.

Version 1:

Decision Letter:

Our ref: NMICROBIOL-24103364A

24th October 2025

Dear Dr. Ogata,

Thank you for submitting your revised manuscript "Giant virus forms a specialized subcellular environment for translation"

(NMICROBIOL-24103364A). I'm very sorry for the delay in getting back to you with a decision, nevertheless, in light of the delay I'm pleased that I'm now able to get back to you with good news. Your work has now been seen by the original referees and their comments are below. The reviewers find that the paper has improved in revision, and therefore we'll be happy in principle to publish it in Nature Microbiology, pending minor revisions to satisfy the referees' final requests and to comply with our editorial and formatting guidelines.

Thank you again for your interest in Nature Microbiology Please do not hesitate to contact me if you have any questions.

Sincerely,

Kyle

Dr. Kyle Frischkorn
(he/him/his)
Senior Editor, Nature Microbiology
Nature Portfolio

Reviewer #1 (Remarks to the Author):

The authors appropriately addressed the reviewer's comments.

Reviewer #2 (Remarks to the Author):

The authors have satisfactorily addressed all my concerns and queries. Thank you.

Reviewer #3 (Remarks to the Author):

I thank the authors for answering the remarks of all 3 reviewers. From my side, I have no further comments.

Reviewer #3 (Remarks on code availability):

The authors provided their entire R workspace. It would be nice if there was explanation on the structure and the different folders/files present there; perhaps an knitted document in html format.

Version 2:

Decision Letter:

8th December 2025

Dear Professor Ogata,

I am pleased to accept your Article "A giant virus forms a specialized subcellular environment within its amoeba host for efficient translation" for publication in Nature Microbiology. Thank you for having chosen to submit your work to us and many congratulations.

You may wish to make your media relations office aware of your accepted publication, in case they consider it appropriate to organize some internal or external publicity. Once your paper has been scheduled you will receive an email confirming the publication details. This is normally 3-4 working days in advance of publication. If you need additional notice of the date and time of publication, please let the production team know when you receive the proof of your article to ensure there is sufficient time to

coordinate. Further information on our embargo policies can be found here:
<https://www.nature.com/authors/policies/embargo.html>

Authors may need to take specific actions to achieve compliance with funder and institutional open access mandates. If your research is supported by a funder that requires immediate open access (e.g. according to <https://www.springernature.com/gp/open-science/plan-s-compliance> Plan S principles or the <https://www.springernature.com/gp/open-science/us-federal-agency-compliance> NIH public access policy) then you should select the gold OA route, and we will direct you to the compliant route where possible. Because authors warrant under our subscription licensing terms that they haven't committed to licensing any version of their article under a licence inconsistent with the terms of our agreement – including the applicable embargo period – publication under the subscription model isn't suitable for authors whose funders require no embargo.

With kind regards,

Kyle

Dr. Kyle Frischkorn
(he/him/his)
Senior Editor, Nature Microbiology
Nature Portfolio

P.S. Click on the following link if you would like to recommend Nature Microbiology to your librarian
<http://www.nature.com/subscriptions/recommend.html#forms>

** Visit the Springer Nature Editorial and Publishing website at http://editorial-jobs.springernature.com?utm_source=ejP_NMicro_email&utm_medium=ejP_NMicro_email&utm_campaign=ejP_NMicro for more information about our career opportunities. If you have any questions please click [here](mailto:editorial.publishing.jobs@springernature.com).

Reviewer #1 (Remarks to the Author):

Zhang et al conducted a study of transfer RNA pools and codon usage for the pair composed of Mimivirus
and its host cell, the amoeba *Acanthamoeba castellanii*.

The discordance between the codon usage of the virus and the amoeba, and the existence of elements of the
translation apparatus, including transfer RNAs encoded by the Mimivirus genome, are intriguing elements.
This definitely deserves to be investigated in depth.

The authors of this manuscript used many and varied techniques and protocols to analyze different aspects
of codon usage and transfer RNA pools within the pair Mimivirus and *Acanthamoeba* sp.

This study is well conducted, ordered in coherent steps. It provides interesting data to decipher some of the
aspects of replication related to the mentioned objectives. However, some would in my opinion deserve to
be further explained and discussed.

We thank the reviewer for his or her careful reading and constructive comments.

Line 231-232: The authors mention similar data obtained in influenza viruses and SARS-CoV-2: "Similar
results have been reported during infections with influenza A virus and coronavirus SARS-CoV-2.". These
two viruses are very different from Mimivirus in terms of size and complexity. What can the authors take
away from such a comparison?

We thank the reviewer for this point. We agree that the Mimivirus and those two viruses have significant
differences. Nonetheless, the shared observations suggest that such a passive way, *i.e.* over-synthesizing
mRNAs and competing for resources, is a shared infection strategy by diverse virus taxa. To avoid any
confusion, we amended the corresponding discussion paragraph as below:

"Viral mRNAs dominated the mRNA pool at the late stages of infection, which is consistent with a previous
study⁸. However, this increase was not perfectly reflected in the footprint proportion (Fig. 1B–C), leading
to a decline in TE of viral mRNAs (Fig. 1H). Low TE during viral infection has been reported for influenza
A virus and SARS-CoV-2^{37,38}. It suggests that overwhelming the host's translation machinery through
massive mRNA synthesis might be a common strategy shared by diverse viral taxa. Consistently, we found
a general increase of A-site empty ribosomes, represented by short footprints along the viral infection stages
(Extended Data Fig. 2P–Q), probably because of the global tRNA or amino acid shortage." (lines 250-257).

Line 239: The authors should further explain this sentence and interesting point: "..., the preference for the
AT-rich genome in APMV may be an adaptive strategy".

We thank the reviewer for the comment. We expanded our discussion about the proposed benefits of an
AT-rich genome as below:

"APMV likely uses a local translation environment to overcome the codon-tRNA mismatch. However, the
reason for the mismatch (*i.e.*, AT-rich genomic feature) still needs an explanation. One hypothesis is that
the AT-rich genome arises from mutational biases, for instance, due to the lack of DNA repair genes, as
proposed for many AT-rich viruses and intracellular microorganisms^{47,48}. However, this scenario might not

apply to APMV, which has a large set of DNA repair genes⁴⁹. Instead, our results brought another possibility
for the origin of the genomic feature. The AT-rich genome might be selectively advantageous by allowing
access to the excessive tRNAs for AU3 codons at the early stage of infection (Fig. 3D). This possibility
may apply to other viruses, as APMV is not the only virus that has a deviated codon usage with its host⁵⁰⁻
⁵². (lines 280-289).

The last sentence of the discussion, lines 245-246: "This translation-associated bottleneck may drive the
continuous acquisition of translation-related genes by the AT-rich genomes of APMV-related viruses",
would also deserve to be further explained, because it describes an intriguing putative mechanism.

We thank the reviewer for the comment. We expanded our discussion as below:

"APMV likely uses a local translation environment to overcome the codon-tRNA mismatch. ... (see
above) ... This possibility may apply to other viruses, as APMV is not the only virus that has a deviated
codon usage with its host⁵⁰⁻⁵². In the case of APMV, this advantage is likely to cease at later infection stages
with high viral mRNA load (Fig. 3D). The viral mRNA load at later infection stages appears to cause a
translation-associated bottleneck, which may be alleviated through the acquisition of translation-related
genes by APMV and related viruses to build favourable translation environments. Viruses in the order
Imitervirales are atypical among viruses by encoding abundant tRNA genes and translation-related proteins
such as aminoacyl-tRNA synthetases (aaRSs)^{53,54}. It was also noted that viral aaRSs were found to be
frequently associated with AT-rich codons⁵³. Similar hypotheses have been proposed for the incorporation
of tRNA genes into other viral genomes⁵⁵⁻⁵⁷" (lines 287-296)

Authors should cite and comment on data from other articles cited below that have described codon usage
and transfer RNA interplay in other viruses, including the points mentioned below:

- Serrano-Solís V, Toscano Soares PE, de Farías ST. Genomic Signatures Among Acanthamoeba polyphaga
Entoorganisms Unveil Evidence of Coevolution. J Mol Evol. 2019 Jan;87(1):7-15. doi: 10.1007/s00239-
018-9877-1. Epub 2018 Nov 20. PMID: 30456441: "The genomic signatures in these entoorganisms of
diverse evolutionary origin are coevolutionarily conserved within an intracellular environment that
provides sanctuary for species of ecological and biomedical relevance. "

- Colson P, Fournous G, Diene SM, Raoult D. Codon usage, amino acid usage, transfer RNA and amino-
acyl-tRNA synthetases in Mimiviruses. Intervirology. 2013;56(6):364-75. doi:10.1159/000354557. Epub
2013 Oct 17. PMID: 24157883: "..., we found that the genes most highly expressed at the beginning of the
Mimivirus replicative cycle have a nucleotide content more adapted to the codon usage in A.castellanii."

- Michely S, Toulza E, Subirana L, John U, Cognat V, Maréchal-Drouard L, Grimsley N, Moreau H,
Piganeau G. Evolution of codon usage in the smallest photosynthetic eukaryotes and their giant viruses.
Genome Biol Evol. 2013;5(5):848-59. doi:10.1093/gbe/evt053. PMID: 23563969; PMCID: PMC3673656:
"We show that 1) Codon usage bias in the host and in the viral genes increases with expression levels and
2) optimal codons use those tRNAs encoded by the most abundant host tRNA genes, supporting the notion
of translational optimization by natural selection. We find evidence that viral tRNA genes complement the
host tRNA pool for those viral amino acids whose host tRNAs are in short supply."

- Sau K, Gupta SK, Sau S, Mandal SC, Ghosh TC. Factors influencing synonymous codon and amino acid
usage biases in Mimivirus. *Biosystems*. 2006 Aug;85(2):107-13. doi: 10.1016/j.biosystems.2005.12.004.
Epub 2006 Jan 24. PMID: 16442213: "It was found that codon usage bias in Mimivirus genes is dictated
both by mutational pressure and translational selection. Evidences show that four factors such as mean
molecular weight (MMW), hydrophathy, aromaticity and cysteine content are mostly responsible for the
variation of amino acid usage in Mimivirus proteins. Based on our observation, we suggest that genes
involved in translation, DNA repair, protein folding, etc., have been laterally transferred to Mimivirus a
long ago from living organisms and with time these genes acquire the codon usage pattern of other
Mimivirus genes under selection pressure."

- Finally, an article on bacteriophages: Bailly-Bechet M, Vergassola M, Rocha E. Causes for the intriguing
presence of tRNAs in phages. *Genome Res*. 2007 Oct;17(10):1486-95. doi: 10.1101/gr.6649807. Epub
2007 Sep 4. PMID: 17785533; PMCID: PMC1987346: "Thus, even though phages use most of the cell's
translation machinery, they can complement it with their own genetic information to attain higher fitness.
These results suggest that similar selection pressures may act upon other cellular essential genes that are
being found in the recently uncovered large viruses."

We thank the reviewer for pointing out these important papers. We cited these works in the Discussion
section in the modified manuscript (numbers 47, 11, 55, 48, and 56).

Line 90 to 95: the proportion of viral mRNAs varies significantly between replicates (62 and 85%, 40 and
81%). How do the authors explain these differences?

We thank the reviewer for this comment. Even with our best efforts, we could not avoid these variations in
biological replicates. These two replicates were prepared in the same week but originated from different
amoeba passages. Sometimes, the amoeba population can change a lot, which might lead to differences in
the infection cycle. To account for potential differences between replicates, we performed our analysis on
each sample individually rather than merging the two replicates. As described in the manuscript, we
observed consistent results in those replicates.

Lines 205-206: "Thus, our data suggest that Mimivirus created a unique environment tailored for viral
mRNA translation at the periphery of the virus factory.": Is this environment physically delimited? How
could it be?

We thank the reviewer for this comment. Our current data do not definitively demonstrate a physical
boundary. A previous study using electron microscopy also only observed that ER surrounded instead of
completely enclosing APMV VF (Mutsafi *et al.*, 2013, *PLOS Pathogens*). Another study isolated APMV
VF without ER (Fridmann-Sirkis *et al.*, 2016, *JVI*). However, they used Triton X-100 and NP-40, both can
destroy the membrane. Further, a recent study demonstrated that APMV VF is made through liquid-liquid
phase separation (Rigou *et al.*, 2025, *BioRxiv*). Thus, currently we think this environment is not physically
delimited.

The authors should comment on whether they think that their observations could also be found for other
viruses for which a virus factory is formed during the replication cycle, referring for example to the
following articles on virus factories:

- Netherton CL, Wileman T. Virus factories, double membrane vesicles and viroplasm generated in animal
cells. *Curr Opin Virol*. 2011 Nov;1(5):381-7. doi: 10.1016/j.coviro.2011.09.008. Epub 2011 Oct 12. PMID:
22440839; PMCID: PMC7102809.

- Novoa RR, Calderita G, Arranz R, Fontana J, Granzow H, Risco C. Virus factories: associations of cell
organelles for viral replication and morphogenesis. *Biol Cell*. 2005 Feb;97(2):147-72. doi:
10.1042/BC20040058. PMID: 15656780; PMCID: PMC7161905.

We thank the reviewer for raising this point. Because ER membranes recruitment and protein re-localization
have been observed for many other viruses, we expect other viruses might have similar observations. We
added a discussion as below:

“In terms of the subcellular organization within the infected cells, VF is a unique intracellular structure that
is formed by many large DNA viruses during infection. For APMV, DNA replication and transcription
occur in the VF^{15,39}, whereas endoplasmic reticulum (ER), ribosomes, and the viral translation initiation
factor SUI1 have been observed outside the VF^{15,17,18}. In this study, we showed that viral mRNAs
surrounded the VF (Fig. 5B) and protein synthesis occurred in the same region (Fig. 5A). Other translation-
related proteins and tRNAs may also be recruited to the same place, which would further increase the local
concentration of components of the translation machinery. Other viruses might also use similar translation
strategies. For example, poxviruses, asfarviruses, and iridoviruses have been reported to induce changes in
cytoskeleton^{40,41}, recruitment and local enrichment of proteins⁴², and even re-localization of organelles
such as mitochondria^{40,43}. Moreover, ER membrane recruitment and protein re-localization have been
observed for RNA viruses as well⁴⁴⁻⁴⁶, indicating that subcellular remodeling is a common feature of viral
infections across diverse virus families.” (lines 267-279)

The authors should comment more on what the implications of their data and their interpretation might be?
Has such a compartment been already observed in other contexts with other microorganisms involved?
Does it exist for other microorganisms and viruses? Is it new?

We added a discussion about how our study is related to previous work and explained its relevance to other
microorganisms.

“In this study, we proposed that the creation of a subcellular translation environment is a key strategy for
APMV to overcome the discrepancy between host and viral codon usages. Virus-induced subcellular
structures have been observed across a wide range of viruses, from prokaryotic to eukaryotic and from RNA
to DNA viruses^{17,58,59}. Although the spatial association between these viral subcellular structures and
translation is largely unknown for other viruses, they may also form a local environment for viral mRNA
translation.” (lines 297-302)

Reviewer #1 (Remarks on code availability):

Possibly the link to the code (10.5281/zenodo.13748128) was not made usable yet as it does not work, or I
did not manage to find the path.

We updated the link to be publicly available, and we updated it to a new version including additional
analysis for review process <https://zenodo.org/records/16555392>.

Reviewer #2 (Remarks to the Author):

The manuscript involves an RNA-seq/Ribo-seq/tRNA-seq study of mimivirus-infected amoeba, backed up
with microscopy including in situ probing of RNA and protein synthesis. Overall the experimental and
bioinformatic work appear to be of a high standard, and the manuscript is clearly written and easy to follow.
However there are some potential issues that need addressing as outlined below.

Major comments:

o I don't think the title "Giant virus creates subcellular environment to overcome codon-tRNA mismatch"
is supported by the data. Yes, APMV creates a subcellular environment - the viral factory (VF), as already
known. But there is no direct evidence that this is "to overcome codon-tRNA mismatch". It was shown that
ribosomes are recruited to the periphery of the VF, and translate viral mRNA there, but no direct evidence
was presented that the local tRNA populations are altered compared to elsewhere in the cell. It's a nice story,
but the evidence presented from the Ribo-seq analysis is indirect. Similarly, the statement on lines 236-238
("APMV alleviates this translation-associated bottleneck by ..." should be written as a conjecture ("APMV
may alleviate ...").

According to the reviewer's suggestion, we amended our title to "Giant virus forms a specialized subcellular
environment for translation" and restructured the Discussion section, which avoided the corresponding
statements.

o One possible alternative explanation might be: "Virus mRNAs are AU rich which means they will have
less RNA structure than the host GC-rich mRNAs. Less structured mRNAs might be translated more easily
by amoeba ribosomes. So maybe lack of structure compensates for lack of codon optimality, rather than the
viral factories altering local tRNA abundances?"

We thank the reviewer for constructive comments. We agree that AU-rich mRNA might be one reason for
a smooth translation elongation due to fewer mRNA secondary structures. However, less-structured
mRNAs alone might not explain why AU3 codons have a lower tRNA accessibility level on amoeba
mRNAs than on virus mRNAs; thus, we still hypothesized a local tRNA abundance model. We included
this part in the discussion as below;

"In terms of translation dynamics and tRNA pool, we found that APMV genes were smoothly translated
without significantly altering the tRNA composition in the host cell (Figs. 2D and 3). This unaltered cellular
tRNA pool aligns with the previous report on vaccinia virus and influenza A virus³⁹. One possible
explanation is that AU-rich mRNAs form fewer secondary structures than GC-rich mRNAs, facilitating a
smooth translation. However, this cannot explain the observed higher tRNA accessibility for AU3-codons
on viral mRNAs than that on AU3-codons on host mRNAs (Fig. 4). These observations imply that a local
tRNA pool could be selectively used for viral mRNAs. APMV may achieve smooth translation by
generating a local translation environment within the cell." (lines 258-266)

o The conclusions are critically dependent on several Ribo-seq analyses, including ribosome occupancy on
codons, fraction of paused codons per gene, and the short/total footprint ratio. Thus it is essential that these
measures are not substantially affected by artifacts. For reasons outlined below, further verification of this
is needed before the results can be properly assessed. Of particular concern is the possibility of ribosome
run-on occurring during sample preparation. As analysed by Hussmann et al 2015 (PMID 26656907),
ribosome run-on can be a real issue, especially when using cycloheximide treatment without flash freezing
(as the authors appear to have done) and this would have the effect of making meaningless any codon-
specific statistics such as the above. Run-on effects can be sensitive to precise details of sample preparation,
besides the cell type/species, so the authors' study may be OK. However it is critically important to
bioinformatically and/or experimentally verify that run-on has not occurred in the authors' datasets. The
analyses outlined in Hussmann et al 2015 could be a good place to start.

We thank the reviewer for raising this point. First of all, the “run-on” was reported 1) only in yeast
(Hussmann *et al.*, 2015, *PLoS Genet*) but not in mammalian cells (Sharma *et al.*, 2021, *Nat Commun*), 2)
only when cells are pretreated with cyclohexamide in the culture (Hussmann *et al.*, 2015, *PLoS Genet*;
Sharma *et al.*, 2021, *Nat Commun*). Generally, current Ribo-Seq (including ours as well) avoids the
pretreatment of cyclohexamide in the culture but applies the drug in the lysis buffer (as “post-lysis
treatment”), as this approach was suggested (Inglia *et al.*, 2012, *Nat Protoc*; McGlincy and Ingolia, 2017,
*Methods*; Mito *et al.*, 2020, *STAR Protoc*). Thus, our protocol does not have any chance of “run-on” artifacts
caused by cyclohexamide pretreatment. We clarified this point in the manuscript.

Nonetheless, accordingly to the reviewer’s suggestion, we tested the possibility of a run-on in our ribosome
profiling data (see the figure below). As conducted in Hussmann *et al.*, 2015, *PLoS Genet*, we depicted the
ribosome footprint occupancy centered on each codon type; if a run-on happens at the specific codon, we
should observe a long trail of ribosome accumulation downstream of the codon (Hussmann *et al.*, 2015,
*PLoS Genet*). However, we did not observe such a trend in mRNAs from amoeba and APMV (see the figure
below). Although we omitted these data from the manuscript due to the display item limitations, we are
happy to add them to supplementary figures if editors and reviewers think it is necessary.

Run-on effects were limited in our dataset.

(A and B) Ribosome occupancy around each codon species was depicted in all the codons (gray, top), AT3 codons (blue, middle), and GC3 codons (yellow, bottom) for amoeba mRNAs (A) and APMV mRNAs (B).

We used 0 hpi samples for amoeba mRNAs and 2 hpi samples for APMV.

o In the analysis of ribosome pausing (lines 130-135), peaks in ribosome protected fragment (RPF) density were defined as pause sites. However, RPF peaks can also result from biases (ligation bias, nuclease bias, PCR bias). More work is needed to determine whether the "mean + 2 standard deviations" threshold is sufficient to select for pause peaks over bias peaks. One possible way forward would be to investigate correlation between RPF peaks and A/P-site codons relative to correlation between RPF peaks and the 5'/3' terminal nucleotides that would have the strongest effect on ligation bias. There is relevant discussion in O'Connor et al 2016 (PMID 27698342). Particularly in the scenario presented in this manuscript, one cannot assume that "bias will average out": because the virus mRNAs are AU-rich and the host mRNAs are GC-rich, they will be subject to very different biases.

We thank the reviewer for raising this point. According to the suggestion, we corrected the bias through multiple methods and then again conducted pausing detection.

Given that the earlier studies (O'Connor *et al.*, 2016, *Nat Commun*; Tunney *et al.*, 2018, *Nat Struct Mol Biol*) indicated biases at the 5' or 3' end of reads, we evaluated those in our data and set out to correct them. Following the approach conducted in Tunney *et al.*, 2018, *Nat Struct Mol Biol*, we performed a multiple linear regression analysis to investigate whether codon types at different positions correlate with A-site occupancy in our dataset. In detail, we analyzed Ribo-Seq data from replicate 2 at 0 hpi and used footprints of 29-30 nt to ensure a clear boundary of ribosome footprints. We applied the following filters to exclude poorly covered genes: (1) average footprint counts per codon < 1, (2) $\geq 80\%$ codons with zero footprint coverage, (3) maximum occupancy per gene > 60, and (4) protein-coding sequences < 100 amino acids.

After filtering, 990 genes were retained for analysis. Centered on the A-site (position 0), flanking codons
from position -5 to +4 were studied. To avoid artifacts from gene boundaries, the window was only
generated starting from the 5th codon downstream of the start codon and ending 5 codons before the stop
codon. We used the statsmodel (v0.14.4) package in Python to perform an Ordinary Least Square (OLS)
regression between these features and the A-site occupancy.

Notably, codon types at not only the P/A-site but also at the -5 position and +4 positions were strong
predictors of A-site occupancy (see panels A-B in the figure below). Codons beginning with “G” at the -5
position consistently showed negative coefficients (see panels A in the figure below). Given that this
position represents the 5' end of the reads, the sequence preferences of enzymes used in the library
preparation may generate the bias as discussed in Tunney
*et al.*, 2018, *Nat Struct Mol Biol*.

To correct this sequencing bias, we developed a codon-correction strategy, modeling the relationship
between A-site occupancy and the identity at the -5 position using linear regression. We tested two
correction approaches.

**Approach 1: Additive correction**

We modeled the original A-site occupancy (O_{orig}) as:

$$295 \quad \quad \quad O_{orig} = \beta_0 + \beta_1 X_1 + \dots + \beta_n X_n$$

Where each X_i is a binary indicator for a codon at the -5 position and β_i is the associated regression
coefficient. The corrected occupancy (O_{corr}) for the window containing the codon X_i at the -5 position
was:

$$300 \quad \quad \quad O_{corr} = O_{orig} - \beta_i$$

In this formula, codons with negative β_i values (*e.g.*, G-starting codons) received a positive correction.

**Approach 2: Multiplicative correction**

We instead assumed a multiplicative relationship:

$$305 \quad \quad \quad O_{corr} = \frac{O_{orig}}{coefficient}$$

Taking the log transformation:

$$308 \quad \quad \quad \log(O_{corr} + 1) = \log(O_{orig} + 1) - \beta_i$$
$$309 \quad \quad \quad O_{corr} = \frac{(O_{orig} + 1)}{\exp(\beta_i)} - 1$$

where β_i is the coefficient estimated from a regression model on log-transformed occupancy.

We applied both methods and set any negative occupancy values to zero. To further analyze the codon
effects, we used a LASSO regression model with 5-fold cross-validation, reserving 30% of the data for
testing. Both correction methods resulted in a decreased correlation between the -5 position with the A-site
occupancy (see panels C-F in the figure below).

**Correlation between codon type at each position and A-site occupancy**

(A, C, and E) Heatmaps showing the correlation between each codon type and A-site occupancy from
 uncorrected data (A), additive correction (C), and multiplicative correction (E).
 (B, D, and F) Bar plots showing the average absolute coefficients of different positions from uncorrected
 data (B), additive correction (D), and multiplicative correction (F).

 Even after those bias corrections, the codon occupancy for each codon type did not change too much (see
 panels G-H in the figure below), suggesting that bias at the -5 position did not affect A-site occupancy
 greatly. Given that the multiplicative approach showed better correction performance, we applied this
 approach to the other samples, using the coefficient learned from the amoeba 0 hpi replicate 2. Ribosome
 pausing on APMV mRNAs was not markedly observed throughout the course of infection (see panel I in
 the figure below).

**Ribosome pauses between before and after correction.**

(G-H) Scatter plots comparing the pausing proportion at A-site using the additive approach (G) and the
 multiplicative approach (H).

(I) Box plots of the fraction of paused codons per gene in the amoeba and APMV genomes at different
 infection stages. The multiplicative approach was used to correct the bias. The median (center line),
 upper/lower quartiles (box limits), 1.5× interquartile range (whiskers), and outlier (points) are shown. The
 *p* values were calculated by the one-sided Wilcoxon rank sum test. ns, not significant; *, *p* <0.05; **, *p*
 <0.01; ***, *p* <0.001; ****, *p* <0.0001.

In summary, the bias generated by library preparation did not impact our assessment of ribosome pausing.
 Although we omitted these data from the manuscript due to the display item limitations, we are happy to
 add them to supplementary figures if editors and reviewers think it is necessary.

o I'm actually surprised the authors saw any of the short (21-22 nt) RPFs without using flash freezing or the
translation inhibitor drugs that others have used to reveal this conformation of the ribosome. There are
difference between e.g. yeast and mammalian cells so it may be that this is a reflection of the different
system (amoeba). Measuring the short/total footprint ratio is a nice idea for aiming to quantify tRNA
accessibility. But I have a few comments

- Sorry if I missed it, but is it specified that this analysis is carried out for the A-site (not P-site) codon?

As the reviewer assumed, we analyzed ribosome occupancy at A-site codons and the ratio of short footprints.
We clarified this in the texts accordingly.

- In the "Short footprint ratio" section in Methods, please specify which read lengths are used for "short"
and for "total" number of footprints. Would the measure be cleaner if it was just 21-22 nt versus 29-30 nt
reads?

According to the reviewer's suggestion, we calculated the ratio of short footprint over long footprint
(Extended Data Fig. 6A-B; see also the figure below). Generally, the same results were obtained, as the
short footprint fraction was employed (Fig. 4A-B).

- In Fig 1C of Cook et al 2022 (PMID 35226596), short and long footprints were detected at different stages
of PRRSV virus infection, but the short/total ratio doesn't seem to be well-correlated with hpi; instead it
seems more likely to be rather sensitive to small differences in sample preparation. The authors' own sample
preparation may be more robust than the Cook et al study in this respect, but they may like to consider
whether possible variation in sample preparation could impact on their data.

We thank the reviewer for a positive evaluation of our data quality. Technically, Cook et al., 2022, eLife
used different brands (Amibion vs. LGC Biosearch Technologies used in this study) of RNase I with
different concentrations. We note that the unit definition of these two brands is quite different. Thus, the
efficiency of trimming could cause a difference in the resolution of two populations of footprints.
Furthermore, they focused on a different virus, which can generate different results of the short/total ratio.

- In nearly all Ribo-seq studies, there is some amount of reads that map to mRNAs but do not derive from
bona fide RPFs, e.g. they might derive from other RNP complexes co-sedimenting with ribosomes. If
nuclease trimming works exceptionally well then an upper limit can be put on non-RPF contamination by
looking at read-length plots (Ext Fig 2A,B) and triplet phasing (proportion of reads mapping to 1st, 2nd and
3rd positions of codons). Even if nuclease trimming doesn't work so well, an upper limit can be put on non-
RPF contamination by looking at 3'UTR:CDS density ratio statistics. For example, if this was 1% for mock
but 15% for 8 hpi, then one would infer that non-RPF contamination was increasing with hpi and might be
15% of the total density at 8 hpi. In the authors' Ext Fig 2A,B, the plots will represent true short RPFs, true
long RPFs, and a background of non-RPF contamination. If this non-RPF contamination increases with hpi
(as is often the case in Ribo-seq studies of other viruses) then it will contribute to the changing short/total
ratio, potentially explaining some or all of the observed increase. Again, as in Ribo-seq studies of other
viruses, the increase in non-RPF contamination at later time points is generally different for host compared
to virus, and - depending on the virus species - can affect host mRNAs more or can affect virus mRNAs
more. It is important that the authors put bounds on how these possibilities might affect the short/total ratio
used in the manuscript. As a start, one could (i) use the 3'UTR density (provided virus and host 3'UTRs are
long enough) to put an upper limit on contamination density at each time point; and (ii) compare virus with
host triplet phasing at each timepoint, separately for the 21-22 nt and 29-30 nt reads. I'd also like to see
"distance from start codon" (like Ext Fig 2C,D) and "distance from stop codon" plots separately for the 21-
22 nt and 29-30 nt reads (e.g. to verify the 21-22 nt reads are RPFs and not miRNAs/siRNAs :-)).

According to the reviewer's suggestion, we added metagene plots for short (21-22 nt) and long (29-30 nt)
footprints around the start codons and stop codons (Extended Data Fig. 2C-J; see also the figure below).
Overall, we observed the clear enrichment of reads at ORFs but not 5' UTR or 3' UTR for both populations
of footprints. This pattern was consistent for both amoeba and APMV mRNAs.

To further test the 3-nt periodicity of the long and short footprints, we conducted a discrete Fourier

transform analysis (Extended Data Fig. 2K; see also the figure below). Despite the mRNA origin (amoeba

and APMV), we found a clear 3-nt periodicity in both long and short footprints along the viral infection.

These last two points could help address some of the above points, but may be impracticable and I put them
here just as suggestions:

o Ideally, the manuscript would include a positive control for what Ribo-seq codon occupancy/pausing
looks like when a tRNA species is in short supply (cf. the Wu et al 2019 paper, PMID 30686592, where
they used 3-amino-1,2,4-triazole (3-AT) treatment to inhibit histidine biosynthesis and deplete histidyl-
tRNA in yeast as a positive control for pausing at A-site His codons). I realise that this may not be a very
practicable question to address for a non-model organism, but it would strengthen claims such as (lines
159-160) "smooth elongation of GC3 codons" implying "no shortage of tRNA supply".

According to the reviewer's suggestion, we conducted a series of experiments with 3-AT on *A. castellanii*.
Given that no prior experiments have examined the effects of 3-AT on *A. castellanii* translation, we assessed
it by *O*-propargyl-puromycin (OP-puro) incorporation. Even with high 3-AT (100 mM, as used in yeasts;
Guydosh and Green, 2014, *Cell*) and titration of incubation time, we did not observe effective suppression
of protein synthesis in amoeba (see panels A-C in the figure below).

Nonetheless, we employed Ribo-Seq with 3-AT (at 100 mM for 3.5 h) (see panel D in the figure
below). Again, the ribosome pauses on the His codons was limited (see panels E in the figure below). Thus,
unfortunately, 3-AT may not be effective in *A. castellanii*.

Characterization of 3-AT treatment in amoeba cells.

- (A) Schematics of OP-puro incorporation assay with 3-AT treatment.
- (B) Gel image for the OP-puro-labeled nascent peptides during 3-AT treatment.
- (C) Quantification of B. The mean (bar), s.d. (error), and individual data (n = 3, points) are shown.
- (D) Length distribution of ribosome footprints.
- (E) Comparison of ribosome occupancy with and without 3-AT treatment. His codons are highlighted in red.

Here, instead, we compared tRNA abundance with codon occupancy, considering that the tRNA pool should directly reflect ribosome traversal rate. As expected, we generally observe a significant negative correlation between ribosome occupancy and tRNA supply (Extended Data Fig. 4C; see also the figure below), confirming that our Ribo-Seq generally reflects ribosome elongation rates in cells.

To validate our detection method, we titrated cutoffs for pausing event detection, including mean + 1 s.d., mean + 2 s.d. (current Fig. 2D), and mean + 3 s.d. (Extended Data Fig. 3A; see the figure below). Irrespective of cutoffs, ribosome pausing on APMV mRNAs was not markedly observed throughout the course of infection.

The manuscript contends that the viral factories (or at least something about virus infection) is responsible for making viral mRNAs efficiently translated despite having codon usage that is very different from host. Is it possible to transfect/overexpress several viral mRNAs in the absence of viral infection, ribosome profile, and assess whether - in contrast to the same mRNAs in virus infection - they exhibit poor translation,

ribosome pausing, etc? I realise this may not be practicable (transfection of a non-model organism,
difficulty getting high enough expression to get enough Ribo-seq coverage, cost).

According to the reviewer's suggestion, we conducted a reporter assay to monitor the effects of AT-rich
and GC-rich sequences in amoeba with or without APMV infection (Extended Data Fig. 7; see also the
figure below).

We employed the established reporter system, which is sensitive to translation elongation effects
(Juszkiewicz and Hegde, 2017, *Mol Cell*; Sundaramoorthy *et al.*, 2017, *Mol Cell*; Han *et al.*, 2020, *Cell*
*Rep*; Ishibashi *et al.*, 2024, *PLoS Biol*). This reporter consists of N-terminal *Renilla* luciferase (Rluc) and
C-terminal firefly luciferase (Fluc) in frame, with the ORF region of the test in the middle. Due to viral 2A
sequences, which induce self-cleavage of peptide bonds during translation (de Felipe *et al.*, 2006, *Trends*
*Biotechnol*), inserted in the middle (Extended Data Fig. 7A; see also the figure below), the separate
individual luciferase proteins allow the ratio of protein synthesis rate upstream and downstream the ORF
region of the test. Here, we investigated the effects of AT-rich ORF originating from TBP of APMV and
GC-rich ORF from actin of amoeba (Extended Data Fig. 7A; see also the figure below).

Apparently, the GC-rich region provided smoother elongation than the AT-rich region, as a higher
Fluc/Rluc ratio is obtained (Extended Data Fig. 7B; see also the figure below). Importantly, the slow
elongation of the AT-rich region was not improved by viral infection (Extended Data Fig. 7B; see also the
figure below). These data should be reasonable, since our reporter mRNA does not possess any
elements/contexts to localize in VF and thus should be translated in the cytosol. Thus, even with our best
efforts (to our best knowledge, this is the very first transfection-based reporter assay in amoeba cells), a
simple reporter assay is technically challenging to recapitulate the translation at VF. Nonetheless, these data
validated that the cytosolic situation is still not suitable for viral mRNA translation and suggested the
requirement of a unique environment like VF.

Notably, although smoother elongation on the GC-rich region upon viral infection (Extended Data Fig. 7B;

see also the figure below), we currently do not have a clear explanation for this phenomenon.

Reviewer #3 (Remarks to the Author): minor

The authors of the manuscript entitled: "Giant virus creates subcellular environmental to overcome codon-
tRNA mismatch" present a compelling case of host cell manipulation by a virus, focusing on the
Mimiviridae infecting Amoebas. In general, the manuscript is very well written and structured, the materials
and methods go in great detail and the figures are of appropriate pixel density and colours to make them
readable and understandable. The authors employed a plethora of molecular and cell biology techniques on
their investigation, presenting a well rounded hypothesis.

As I already mentioned, the manuscript is quite well written and structured. However, there are some terms
and concepts that are introduced at a rather late stage. A prime example I noticed was the definition of the
"footprints" and their significance, that are already mentioned in the introduction and results, but are only
defined later. I strongly suggest that the authors introduce the term already in the introduction.

We thank the reviewer for the positive comments. We have modified it accordingly as below;
"Ribo-Seq captures ribosome-protected fragments of mRNA generated by RNase treatment, so-called
"ribosome footprints", thereby enabling high-resolution measurements of translation dynamics." (lines: 65-
67).

In the materials and methods it would be beneficial to write how much raw sequencing data was produced
524 per sample or per investigation (e.g. line 288: "..pair-end read option, and a total yield of XX gigabases).
All other methods are described in great extent.

According to the reviewer's suggestion, we added Extended Data Table 1 (see also the table below).

**Number of reads yielded in each sample.**

	Replicate 1 (GB)	Replicate 2 (GB)	Library
0 hpi	1.27	1.65	Ribo-Seq
2 hpi	1.42	1.20	
4 hpi	1.57	0.98	
8 hpi	1.26	1.74	
0 hpi	8.16	7.22	RNA-Seq
2 hpi	8.88	7.11	
4 hpi	8.66	8.68	
8 hpi	6.69	7.24	
0 hpi	2.11	1.99	mim-tRNA-Seq
2 hpi	2.61	2.19	
4 hpi	1.98	1.93	
8 hpi	2.99	2.68	

My major comments are on the conclusions and on the Boxplots. If we start with the boxplots (especially
Figures 1 and 2) it seems that there is something going on with the statistics. In particular, if we take fig
1D, 1E and 1H as examples, there is suggested to be a very strong statistical significance in the differences
between Amoeba and APMV, even though it does not look like from the distribution of observations. Since
the code is not available yet, is this an artifact of the analysis that puts too much weight on the actual number
of observations (Amoeba: 10575, APMB 955) which would overshadow any other effect? The same applies
for Figure 2. Fig 3A and 3B have similar numbers of observations between host and virus and there there
is no statistically significant difference, while for Fig 3D there is an effect, that again may be affected just
by the number of observations. Code availability to the reviewers would be able to answer this.

We apologize for the inaccessibility of our code deposit in Zenodo. We have updated the link to be publicly
available and updated it to a new version including additional analysis for review process
(<https://zenodo.org/records/16555392>).

From a statistical standpoint, when the sample size is very large, even subtle differences between
groups can become statistically significant, as the tests gain more power to detect small effects. Here, we
are using the Wilcoxon rank sum test, which works when the number of samples is over 10. For the unbiased
and fair comparison, differences in the mRNA numbers in each category are unavoidable.

Nonetheless, to address the reviewer's concern, we subgrouped mRNAs from each category to
maintain the same number of observations (see figure below). Here, we randomly selected 300 mRNAs
from amoeba and APMV mRNAs, respectively, to repeat the statistical analysis in Fig. 1D, 1E, 1H, and
2D. Then, we repeated this subsampling and tested 100 times (see the figure below). Overall, this repetitive
subsampling analysis was aligned well with the main figures.

For Ribo-Seq and RNA-Seq TPM (corresponding to Fig. 1D and 1E; see panels A and B below),
every subsample showed significant differences ($p < 0.05$). For TE (corresponding to Fig. 1H; see panel C
below), most subsampling tests were non-significant at 2 hpi ($p \geq 0.05$: 83/100 in replicate 1 and 89/100 in
replicate 2) and then, the number of subsampling tests showing $p < 0.05$ increased at 4 hpi (93/100 in
replicate 1 and 90/100 in replicate 2) and at 8 hpi as well; again this observation recapitulate the data in the
main text.

For pausing fraction in amoeba and APMV mRNAs (corresponding to Fig. 2D; see panel D below),
the number of subsamples showing a $p < 0.05$ was high (see the table below). Only the sample of replicate
1 at 8 hpi showed an exception; a small number of subsampling results showed a $p < 0.05$ (*i.e.*, not
significant in 11/100 subsamples), indicating a non-significant relationship. These results were the same as
those in our main text, which only replicate 1-8hpi is non-significant, and all others are significant.
Although we omitted these data from the manuscript due to the display item limitations, we are happy to
add them to supplementary figures if editors and reviewers think it necessary.

Subsampling and repeated statistical tests

(A-D) Density plot showing the distribution of p -value from 100 times subsampling from (A) Ribo-Seq

TPM, (B) RNA-Seq TPM, (C) TE, and (D) fraction of paused codons per gene. The red dashed line

represents $-\log_{10}(0.05)$.

My last remark is regarding the suggestion of a microcompartment in the host's cell that allows the virus to

snatch all available AU3-related tRNAs. I was of course happy to read that the authors suggest an evolution

of the virus to make use of the high GC preference of the host, but from Extended Data Fig.1A there seem

to be virtually no *Amoeba* codons that have GC content below 50%. When I read this figure I see basically

zero competition between host and virus, so the fact that ribosomes are close to the VF and that viral mRNA

translation proceeds happily may just be due to the fact the host does not need these tRNAs at all so they

naturally are found closer to the VF upon infection. The authors also suggest that synthesis of viral proteins

is limited by tRNA availability because of the availability of viral mRNAs and the ration of AT-rich to GC-

rich tRNAs in the host. For that to be viable there would need to be data showing either some kind of

transcription regulation in viruses, translation kinetics in the presence of abundant tRNAs or an effect of

different concentrations of the right tRNAs and the number of viral particles.

We thank the reviewer for the constructive comment. We consider that the reviewer and we probably share

the same idea about the viral avoidance of the resource competition with the host. Indeed, this encouraged

589 us to conduct the analysis of Fig. 3D, through the balance score. This analysis revealed an excess of AU3-
590 tRNAs, which may be exploited by the virus. This analysis however clearly showed that viral infection led
to the shortage of tRNAs for AU3 codons over time, considering host and viral mRNA abundance.
Apparently, this was due to the over accumulation of viral mRNAs (Fig. 1B and 1D), given the tRNA pool
is stable (Fig. 3C). Nonetheless, our Ribo-Seq data showed that ribosome traversal was smooth (Fig. 2D).

This discrepancy encourages us to hypothesize that the local environments may facilitate viral
translation to overcome high tRNA demands. Our microscopic analysis indicated that the VF may serve as
a place to support the viral protein synthesis (Fig. 5).

To clarify our interpretation based on this consideration, we modified a part of discussion section
as follows:

“APMV likely uses a local translation environment to overcome the codon-tRNA mismatch. However, the
reason for the mismatch (i.e., AT-rich genomic feature) still needs an explanation. One hypothesis is that
the AT-rich genome arises from mutational biases, for instance, due to the lack of DNA repair genes, as
proposed for many AT-rich viruses and intracellular microorganisms^{47,48}. However, this scenario might not
apply to APMV, which has a large set of DNA repair genes⁴⁹. Instead, our results brought another possibility
for the origin of the genomic feature. The AT-rich genome might be selectively advantageous by allowing
access to the excessive tRNAs for AU3 codons at the early stage of infection (Fig. 3D). This possibility
may apply to other viruses, as APMV is not the only virus that has a deviated codon usage with its host⁵⁰⁻
⁵². In the case of APMV, this advantage is likely to cease at later infection stages with high viral mRNA
load (Fig. 3D). The viral mRNA load at later infection stages appears to cause a translation-associated
bottleneck, which may be alleviated through the acquisition of translation-related genes by APMV and
related viruses to build favourable translation environments. Viruses in the order Imitervirales are atypical
among viruses by encoding abundant tRNA genes and translation-related proteins such as aminoacyl-tRNA
synthetases (aaRSs)^{53,54}. It was also noted that viral aaRSs were found to be frequently associated with AT-
rich codons⁵³. Similar hypotheses have been proposed for the incorporation of tRNA genes into other viral
genomes⁵⁵⁻⁵⁷” (lines 280-296)

Furthermore, to show that the cytosolic tRNA availability affects the translation speed for AU-rich and GC-
rich mRNAs, we performed additional analyses and experiments.

As described above for the response to Reviewer #2, we compared tRNA abundance with codon occupancy,
considering that the tRNA pool should directly reflect ribosome traversal rate. As expected, we generally
observe a significant negative correlation between ribosome occupancy and tRNA supply (Extended Data
Fig. 4C; see also the figure below), suggesting that tRNA abundance affects the translation elongation speed.

Further, we set out a reporter assay to monitor the effects of AT-rich and GC-rich sequences in amoeba
 with or without APMV infection (Extended Data Fig. 7; see also the figure below). Apparently, the
 reporter's GC-rich region resulted in a higher Fluc/Rluc ratio than the AT-rich region, suggesting a
 smoother elongation for GC-rich mRNAs (Extended Data Fig. 7B; see also the figure below). The slow
 elongation of the AT-rich region was not improved, while a smoother elongation of the GC-rich region was
 maintained upon viral infection (Extended Data Fig. 7B; see also the figure below). These observations
 suggest that the cytosolic environment performs a smoother translation of GC-rich mRNAs, further
 supporting the requirement of a unique environment for viral protein synthesis.

Reviewer #3 (Remarks on code availability):

The Zenodo archive that hosts the code is not yet publicly available. The same applies for the NCBI
repositories that host the raw sequencing data.

We updated the link for initial submission (<https://doi.org/10.5281/zenodo.13901452>) to be publicly
available. We updated it to a new version including additional analysis for review process
(<https://zenodo.org/records/16555392>).